# TOWARDS LAYER-WISE PERSONALIZED FEDERATED LEARNING: ADAPTIVE LAYER DISENTANGLEMENT VIA CONFLICTING GRADIENTS

## ABSTRACT

In personalized Federated Learning (pFL), high data heterogeneity can cause significant gradient divergence across devices, adversely affecting the learning process. This divergence, especially when gradients from different users form an obtuse angle during aggregation, can negate progress, leading to severe weight and gradient update degradation. To address this issue, we introduce a new approach to pFL design, namely Federated Learning with Layer-wise Aggregation via Gradient Analysis (FedLAG), utilizing the concept of gradient conflict at the layer level. Specifically, when layer-wise gradients of different clients form acute angles, those gradients align in the same direction, enabling updates across different clients toward identifying client-invariant features. Conversely, when layer-wise gradient pairs make create obtuse angles, the layers tend to focus on client-specific tasks. In hindsights, FedLAG assigns layers for personalization based on the extent of layer-wise gradient conflicts. Specifically, layers with gradient conflicts are excluded from the global aggregation process. The theoretical evaluation demonstrates that when integrated into other pFL baselines, FedLAG enhances pFL performance by a certain margin. Therefore, our proposed method achieves superior convergence behavior compared with other baselines. Extensive experiments show that our FedLAG outperforms several state-of-the-art methods and can be easily incorporated with many existing methods to further enhance performance.

## 1 INTRODUCTION

The challenge of non-independent and non-identically distributed (non-IID) data significantly impacts personalized Federated Learning (pFL). Addressing the aforementioned problem, numerous researchers have delved deeply into various directions, e.g., 1) consisting adding regularization (T. Dinh et al., 2020; Li et al., 2020), 2) pseudo representations generation (Zhu et al., 2021; Zhang et al., 2022), 3) on-client data pre-processing (Huang et al., 2022; 2024a), 4) gradient update modification (Reddi et al., 2021; Wang et al., 2020; Sun et al., 2023; Huang et al., 2024b), 5) cross-client feature alignments (Zhang et al., 2023a; Dinh et al., 2022), and 6) adaptive global update by leveraging on-server gradients (Jhunjhunwala et al., 2023; Panchal et al., 2023). These approaches are *orthogonal* in nature, enabling their *combined integration to achieve further performance enhancement*.

Lately, model layer disentanglement has emerged as a promising approach to further enhancing the performance of pFL (Oh et al., 2022; Collins et al., 2021; Chen & Chao, 2022; Xu et al., 2023). This approach involves the disentanglement of local model into two distinct components: global aggregation layers (GAL) and personalized layers (PL). GAL manages common tasks among all users, while PL handles the specific tasks according to each user. However, existing methods require extensive fine-tuning to determine which layers should be used for global aggregation and personalization. Consequently, the optimal selection for layer disentanglement remains an open question.

Addressing this challenge, we consider the gradient conflict (Yu et al., 2020b) in multi-task learning (MTL). The principle of gradient conflict posits that if the angle between the gradients from two users is less than $\pi/2$, the gradients are aligned in the *same direction*. Consequently, the gradient progress of each user does not negatively impact the progress of the others (see Fig. 1). By utilizing gradient conflict, it becomes possible to monitor user interactions directly from the server without local data accessibility, i.e., the extent to which one user's progress may adversely affect others.

Moreover, when analyzing gradient conflict at the layer level, we discovered that layer-wise gradient conflicts do not align in a structured way. Specifically, the initial layers tend to exhibit low gradient conflicts, indicating their role in learning generic features, whereas the deeper layers show high gradient conflicts, reflecting their role in learning more personalized tasks (see Section 3). This observation is also aligned with the phenomenon in MTL (Shi et al., 2023).

Building upon the rationale of gradient conflict and the observation of layer-wise gradient conflict among users in FL, we hypothesize that leveraging layer-wise gradient conflict enables optimal selection for layer disentanglement in pFL. To this end, we propose Federated Learning with Layer-wise Aggregation via Gradient Analysis (FedLAG). Specifically, FedLAG utilizes gradients received from local users to analyze and determine the optimal approach for layer disentanglement via layer-wise gradient conflict. The benefits of FedLAG over current pFL include:

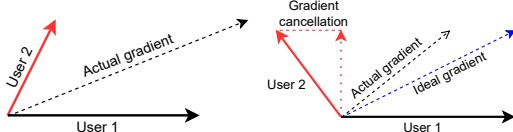

Figure 1: The issue of gradient conflicts, we denote ideal gradient as the aggregated gradient when component gradients do not make any conflicts, i.e., when the conflicted gradient with angle more than $\pi/2$ is projected into $\pi/2$ as mentioned in (Yu et al., 2020b).

- Ability to evaluate the performance of deep learning models at the layer level.
- Ability to analyze statistics of local user data via users' gradients.
- Efficiently reduce communication overhead by utilizing the gradients.

Our contributions can be summarized as follows: 1) We develop an algorithm, namely Federated Learning with Layer-wise Aggregation via Gradient analysis (FedLAG), which can automatically differentiate between the GAL and PL layers without requiring extensive tuning. 2) We give a theoretical evaluation and present evidence showcasing the convergence of the proposed algorithm. We illustrate its sustained superior convergence compared to conventional FL algorithms without the incorporation of FedLAG. 3) We conduct extensive experimental evaluations to show the superiority of the proposed algorithm over conventional FL algorithms.

## 2 PROBLEM FORMULATION & PRELIMINARIES

### 2.1 NOTATIONS

We use $\mathrm{span}(g_1^{(r)}, \ldots, g_J^{(r)})$ as the subspaces that are contained by all vectors $\{g_1^{(r)}, \ldots, g_J^{(r)}\}$. We consider an FL system comprising a set of users denoted by $\mathcal{U} = \{u | u = 1, 2, \ldots, U\}$. Each user gains access to its local data, which remains inaccessible to others. Specifically, the data collected by the $u$-th user can be represented by $\mathcal{D}_u \in \mathbb{R}^{N_u}$, where $N_u$ denotes the number of data instances for user $u$. The entire dataset available across all users can be denoted by $\mathcal{D} = \bigcup_{u \in \mathcal{U}} \mathcal{D}_u$, and we have $N = \sum_{u=1}^{U} N_u$. We use the term $E$ to represent the number of local epochs. We abuse the notations $w_u^{(r)}$ and $w_g^{(r)}$ to refer to the local model of user $u$ and the global model at round $r$, respectively.

### 2.2 PROBLEM SETUP

During FL training, users iteratively conduct local training and communicate with the server for model updating. To be specific, our FL concept works as follows:

**Local Updates.** In each round $r$, users' local models are updated by the global model, i.e., $w_u^{(r)} \leftarrow w_g^{(r)}$, $\forall u$. Then, users conduct local training in parallel. We assume that the local models parallelly solve the empirical loss over data distribution of the user $u$. For instance, $\mathcal{L}(w_u^{(r,e)}) = \mathbb{E}_{B_k \in \mathcal{D}_u}\left[\ell(B_k, w_u^{(r,e)})\right]$, where $B_k$ is the $k^{\text{th}}$ mini-batch sampled from $\mathcal{D}_u$, $k$ represents the data batch index, and $e$ as the local epoch. In each training epoch, the $\mathcal{L}(w_u^{(r,e)})$ is computed by averaging over the batch-wise loss computation $\ell(B_k, w_u^{(r,e)})$ over the all batches $B_k$, $\forall k \in \{1, \ldots, K\}$. Subsequently, at every FL communication round, users update with a local

learning rate $\eta$ as $w_u^{(r,E)} \leftarrow w_u^{(r)} - \sum_{e=0}^{E-1} \eta \nabla \mathcal{L} \left( w_u^{(r,e)} \right)$, where $\mathcal{L}, \nabla\mathcal{L}$ are the empirical loss function and its gradient, respectively.

**Global Aggregation.** After local training, the server aggregates the local models $w_g^{(r+1)} \leftarrow \frac{1}{U} \sum_{u=1}^{U} w_u^{(r,E)}$ where $N_u$ is the number of samples from user $u$ and $N = \sum_{u=1}^{U} N_u$. The server disseminates the global parameters $w_g^{(r+1)}$ to the local users chosen for the following round $r+1$.

**Layer Disentanglement.** Assume user models $w_u^{(r)}$ consisting of $L$ layers, i.e., $w_u^{(r)} = \{\theta_{l,u}^{(r)}\}_{l=1}^{L}$, where $\theta_{l,u}^{(r)}$ represents the parameter weights at $l^{th}$ layer. Our objective is to disentangle the $L$ layers into GAL and PL. The GAL is defined as $\theta_{s,u}^{(r)} = \{\theta_{l,u}^{(r)} | \forall l \in \mathbb{L}_g\}$. On the other hand, PL is defined as $\theta_{p,u}^{(r)} = \{\theta_{l,u}^{(r)} | \forall l \in \mathbb{L}_p\}$. Here, $\mathbb{L}_g$ and $\mathbb{L}_p$ represent the layers assigned to GAL and PL, respectively. We have $w_u^{(r)} = \{\theta_{l,u}^{(r)}\}_{l=1}^{L} = \{\theta_{s,u}^{(r)}, \theta_{p,u}^{(r)}\}$. Similarly, we denote $w_g^{(r)} = \{\theta_{l,g}^{(r)}\}_{l=1}^{L} = \{\theta_{s,g}^{(r)}, \theta_{p,g}^{(r)}\}$.

## 2.3 NEGATIVE TRANSFER AND GRADIENT CONFLICTS IN MULTI-TASK LEARNING

A major challenge for MTL is *negative transfer*, which refers to the performance drop on a task caused by the learning from other tasks, resulting in degradation in overall performance. A rationale of this phenomenon is the *conflicting gradients* (Yu et al., 2020a). Specifically, gradients from different tasks may point in different directions so that directly optimizing the average loss is detrimental to a specific task's performance. Denote $g_i = \nabla\mathcal{L}(w_i)$ as the gradient of task $i$, and $\phi_{ij}$ as the angle between two task gradients $g_i$ and $g_j$, we have

**Definition 2.1** (Conflicting gradients (Yu et al., 2020a)). Given two gradients $g_i, g_j$ $(i \neq j)$, and $\cos\phi_{ij} = \frac{g_i \cdot g_j}{\|g_i\|\|g_j\|}$ is the cosine between two vectors. $g_i$ and $g_j$ $(i \neq j)$ are said to be conflicting with each other if $\cos\phi_{ij} < 0$.

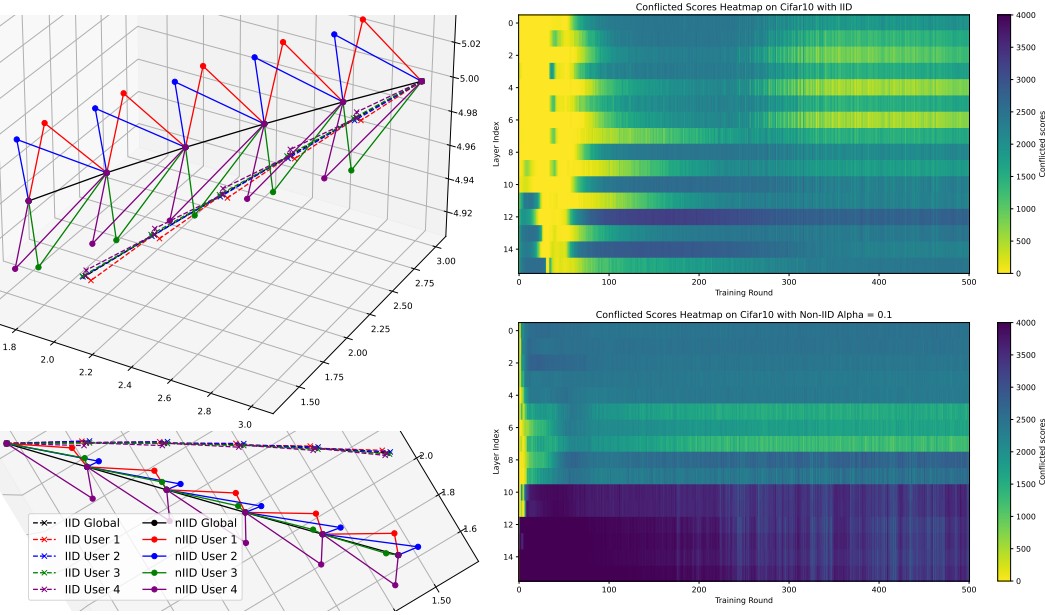

(a) Illustration of gradient conflict with toy dataset.   (b) Layer-wise gradient conflict on Cifar-10.

Figure 2: Illustration of gradient conflicts on 2D toy dataset with 1-layer 3-parameter network (left), and layer-wise gradient conflict of FedAvg during the training (right).

In (Shi et al., 2023), the concept of conflicting gradients is extended to the layer level.

**Definition 2.2** (Layer-wise Conflicting Gradients Shi et al. (2023)). The gradients $g_{l,i}, g_{l,j}$ $(i \neq j)$ of layer $l$ are said to be conflicting with each other if $\cos\phi_{ij}^l < 0$.

The definition suggests that gradient conflicts within a model can be analyzed at the layer level, offering a foundation for exploring and determining optimal strategies for layer disentanglement.

## 3 VALIDATION OF LAYER-WISE GRADIENT CONFLICTS IN FL

We argue that the aggregation of layers with layer-wise gradient conflicts is detrimental in FL. To validate our arguments, we conduct two experiments to answer two questions:

**Question 3.1.** Is gradient conflicts appear among users in federated learning?

To answer the question 3.1, we conduct the experiments on 2-D toy dataset (the details of the toy dataset is reported in Appendix G). The result is visualized Fig. 2a. In the non-IID setting, the divergences between two pairs of gradients (e.g., $g_1^{(r)}$ vs. $g_3^{(r)}$ and $g_2^{(r)}$ vs. $g_4^{(r)}$) result in the divergence of the FL process (Fig. 2a, top). Due to gradient conflicts (angles exceeding $\pi/2$), the model diverges along two hyper-spaces, $\mathrm{span}(g_1^{(r)}, g_3^{(r)})$ and $\mathrm{span}(g_2^{(r)}, g_4^{(r)})$ (Fig. 2a, bottom). In contrast, under IID settings, the gradients form smaller angles and align toward consistent directions. Consequently, the model progresses more directly toward the optimal solution. As noted in Appendix G, the optimal solution is located at $y = 0$, implying that $w_1 = 0$, $w_2 \neq 0$, and $b = 0$.

**Question 3.2.** Is the gradient conflict distribution aligned in an ascending way (i.e., from the input layers to the output layers) during the training of federated learning?

To answer the question 3.2, we conduct the experiments on Cifar-10 dataset and visualize in Fig. 2b. As observed in the figure, the distribution of gradient conflicts does not correspond with the assumptions regarding layer disentanglement found in current SOTA, e.g., Oh et al. (2022), Collins et al. (2021). Specifically, the density of conflicting gradients among clients does not progressively increase from the input layer to the output layer, as these approaches apply to design the layer disentanglement (i.e., personalized layers are assigned at the very last layers).

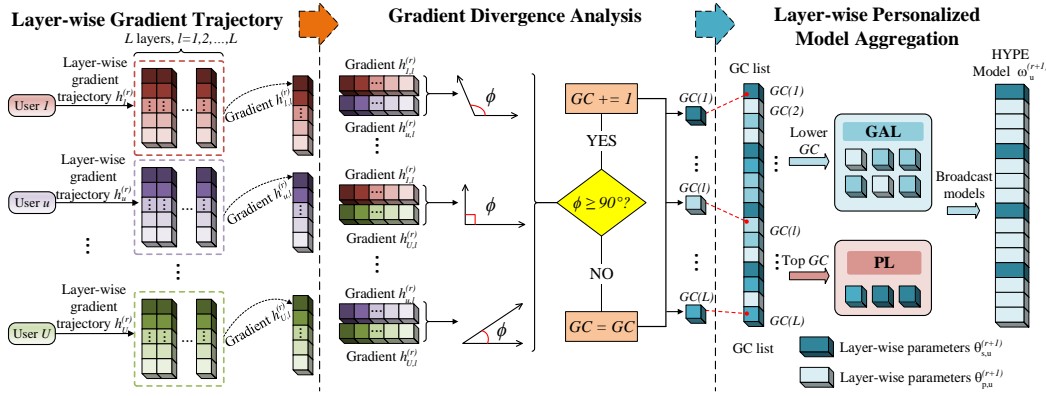

Figure 3: The FedLAG architecture. First, calculate the previous layer-wise gradient $h_u^{(r-1)}$ using received and stored models. Second, measure angles between pairs of gradient vectors, considering angles above 90 degrees as conflicts. Thirdly, the $GC_\epsilon(l)$ (in Definition 4.1) score for layer $l$ increases by $1$ for each conflicted pair. The $k$ layers with highest $GC_\epsilon(l)$ are assigned to PL. The GAL and PL are used to assign the personalized and global aggregated layers.

## 4 METHODOLOGY

Building upon the validations presented in Section 3, we introduce FedLAG, a method that exploits layer-wise gradient conflicts to identify the optimal selection for disentangling personalized and generic layers. The process of FedLAG is briefly illustrated as in Figure 3. The core contribution of FedLAG is its ability to utilize users' gradients as a proxy for their behavior on the server, enabling an analysis of user relationships *without direct access to local datasets*. By taking user gradients as

input, FedLAG applies the gradient divergence analysis (GDA) to examine gradients at the layer level (see Section 4.1). Upon performing GDA, the $GC_\xi(l)$ score is computed, which quantifies the degree to which a given layer should be personalized. Consequently, we can leverage $GC_\xi(l)$ to adaptively disentangle generic and personalized layers (see Section 4.2).

## 4.1 GRADIENT DIVERGENCE ANALYSIS

**Layer-wise gradient.** To implement the GDA, we calculate the gradient of each user $u$ on round $r$:

$$h_u^{(r)} = \sum_{e=0}^{E-1} \eta \nabla \mathcal{L}(w_u^{(r,e)}) = w_u^{(r,E)} - w_u^{(r)}, \tag{1}$$

where the layer-wise gradient $h_{l,u}^{(r)}$ of $h_u^{(r)}$ can be represented as

$$h_{l,u}^{(r)} = \sum_{e=0}^{E-1} \eta \nabla \mathcal{L}(\theta_{l,u}^{(r,e)}) = \theta_{l,u}^{(r,E)} - \theta_{l,u}^{(r)}. \tag{2}$$

Here, $h_u^{(r)}$ and $h_{l,u}^{(r)}$ denote the gradient of $w_u^{(r)}$ and the $l^{th}$ layer of $w_u^{(r)}$, respectively. We further define $h_u^{(r)} = [h_{1,u}^{(r)}, h_{2,u}^{(r)}, \ldots, h_{L,u}^{(r)}]$ where $h_{l,u}^{(r)}$ is a $l$-th layer-wise gradient of user $u$. The gradient is calculated by utilizing the models from the previous round in conjunction with the recently received model. As a result, FedLAG is both *communication-efficient* and capable of *information aggregation*.

**Layer-wise Gradient Divergence Analysis.** To evaluate the layer-wise gradient divergence between two users, we introduce $\phi_l^{(r)}(u, v)$ to signify the angle between $h_{l,u}^{(r)}$ and $h_{l,v}^{(r)}$. Based on Definition 2.1, we define gradient conflict of layer $l$ in the round $r$ with hyper-parameter $\xi$ (w.r.t $-1 < \xi \leq 0$) as the *layer-wise gradient conflict score*, denoted by $GC_\xi(l)$.

**Definition 4.1** ($GC_\xi(l)$ score). Given the threshold $\xi$, the $GC_\xi(l)$ of the $l$-th layer is calculated as the number of distinct user pairs $(u, v)$ (where $u \neq v$) that satisfy $\cos \phi_l^{(r)}(u, v) < \xi$. For instance,

$$GC_\xi(l) = \frac{1}{2} \sum_{u=1}^{U} \sum_{v=1, v \neq u}^{U} \mathbb{I}_{u,v}, \quad \text{s.t.} \quad \mathbb{I}_{u,v} = \begin{cases} 1, & \text{if } \cos \phi_l^{(r)}(u, v) < \xi, \\ 0, & \text{otherwise,} \end{cases} \tag{3}$$

where $\cos \phi_l^{(r)}(u, v) = \cos(h_{l,u}^{(r)}, h_{l,v}^{(r)})$, $\forall v, u \in U$; $\xi$ denotes the extent of conflict severity. To elaborate, by setting a smaller value for $\xi$, the angle between the two vectors becomes more obtuse.

Consequently, the Definition 4.1 allows us to focus primarily on the count of more prominent conflicts. Subsequently, $GC_\xi(l)$ acts as an indicator for conflicting gradients across various severity levels within the layers. If $GC_\xi(l)$ takes on the value $\binom{U}{2}$, it implies that for any two users, there is a conflict in their gradients w.r.t the $l$-th layer. By computing these layer-wise conflict scores, we can pinpoint the layers where conflicts occur most frequently.

We describe our method to find personalized layers $\mathbb{L}_p$ in Algorithm 2. First, we calculate $h_{l,u}^{(r)}$ for each layer by Eq. equation 2. Because we utilize the users' model parameters to measure users' gradient trajectories $h_u^{(r)}$. *No additional communication cost is incurred for the execution of Algorithm 2*. Afterwards, we calculate cosine $\cos \phi_l^{(r)}(u, v)$ by Definition 2.1 and $GC_\xi(l)$ score via the Definition 4.1. Finally, we find the $k$ layers with the highest scores and assign their index to $\mathbb{L}_p$.

## 4.2 LAYER-WISE PERSONALIZED MODEL AGGREGATION

To achieve adaptive layer disentanglement, we base our approach on two key principles: (1) *maintain the global aggregation of FL on non-conflict layers*, and (2) *motivate the personalized learning on conflict layers*. Specifically, when a layer is significantly affected by gradient conflict, we transform it into a personalized layer to prevent negative transfer resulting from the aggregation process at the global server. To this end, rather than broadcasting the entire model to the local users, we restrict the local model update to the global layer (i.e., layers assigned to GAL):

$$\theta_{l,u}^{(r+1)} = \frac{1}{U} \sum_{u=1}^{U} \theta_{l,u}^{(r,E)}, \quad \text{if } l \in \mathbb{L}_g. \tag{4}$$

To motivate the personalized learning, the users do not update the conflict layers:

$$\theta_{l,u}^{(r+1)} = \theta_{l,u}^{(r,E)}, \qquad \text{if } l \in \mathbb{L}_p. \tag{5}$$

Here, $\mathbb{L}_p$ represents the layers that suffer from the gradient conflict and need to be converted into personalized layers. The detailed description of our method is demonstrated in Algorithm 1. Compared to the conventional pFL, FedLAG only alters the aggregation process on the global server, leaving the local training unaffected. Consequently, our technique is applicable to various existing FL or pFL approaches.

## 5 THEORETICAL ANALYSIS

In this section, we discuss the convergence of our algorithm. Our theoretical analysis aims to show the improvement in terms of upper bound reduction for the convergence upper boundary.

### 5.1 LAYER-WISE LOSS IMPROVEMENT

To show the robustness and prove the convergence of FedLAG, we first want to analyze the performance improvement when using our LAG algorithm in FL.

**Lemma 5.1** (Personalization Improvement). *Each user $u$ achieve an improvement in loss when using FedLAG over the vanilla FL approach as follows:*

$$\mathcal{L}\Big(w_{LAG,u}^{(r)}\Big) - \mathcal{L}\Big(w_{VFL,u}^{(r)}\Big) = -\eta \frac{1}{U} \sum_{v=1}^{U} \sum_{l \in \mathbb{L}_p} \|h_{l,u}^{(r-1)}\| \times (\|h_{l,u}^{(r-1)}\| - \cos \Phi_{u,v}^{(l)} \|h_{l,v}^{(r-1)}\|) < 0,$$

**Lemma 5.2** (Generalization Improvement). *For any sufficiently small learning rate $\eta$, the following holds:*

$$\mathcal{L}\Big(w_{LAG,g}^{(r)}\Big) - \mathcal{L}\Big(w_{VFL,g}^{(r)}\Big) = -\eta \frac{1}{U^2} \sum_{u=1}^{U} \sum_{v=1}^{U} \sum_{l \in \mathbb{L}_p} \|h_{l,u}^{(r-1)}\| \times (\|h_{l,u}^{(r-1)}\| - \cos \Phi_{u,v}^{(l)} \|h_{l,v}^{(r-1)}\|) < 0,$$

*where $w_{LAG,g}^{(r)} = \{\theta_{s,g}, \theta_{p,g}\}$ and $w_{VFL,g}^{(r)} = \{\theta_{s,u}, \theta_{p,u}\}$ stand for the FL with and without the integration of FedLAG, respectively.*

According to Lemma 5.2, at each round $r$, the loss function of the layer-wise pFL model consistently surpasses that of the vanilla pFL model by a specific margin. This enhancement is directly proportional to both the layer-wise gradient norm from the previous round $r-1$ and the layer-wise angle between pairs of users in the FL system. Consequently, the proof establishes that the application of FedLAG is viable by relying on the previous gradients for estimating gradient conflicts, *eliminating the need for using the upcoming local gradients*, which is unavailable at the global server. This approach avoids communication overheads associated with exchanging information among local users. Furthermore, the improvement remains constant in each round upon integrating the LAG, in contrast to the FL algorithm lacking LAG integration. The proof of Lemma 5.2 is provided in Appendix H.4.

### 5.2 CONVERGENCE ANALYSIS

We use Assumptions H.1, H.2, H.4, H.3 and have the following theorem:

**Theorem 5.3.** *Assuming users compute full-batch gradient with full participation and $\frac{1}{2\sqrt{6}E^2L} < \eta < \frac{1}{6EL}$, the series $\{w^{(r)}\}$ generated by FedLAG satisfy:*

$$\mathcal{L}(w^{(R)}) - \mathcal{L}(w^*) \leq \mathcal{O}\Big(\frac{\|w_g^{(0)} - w^*\|^2}{R\eta E}\Big) + \mathcal{O}\Big(\eta \sigma_*^2\Big) + \mathcal{O}\Big(\eta^2 E(E-1)L\sigma_*^2\Big) \tag{6}$$

$$- \mathcal{O}\Big(A \sum_{u=1}^{U} \sum_{v=1}^{U} \sum_{l \in \mathbb{L}_p} \|h_{l,u}^{(r)}\|(\|h_{l,u}^{(r)}\| - \cos \Phi_{u,v}^{(l)} \|h_{l,v}^{(r)}\|)\Big),$$

*where $w^* \triangleq \arg\min_w \mathcal{L}(w)$ is the optimal global model, and $A = \Big(24E^2(E-1)\eta^4L^2 - \eta^2E\Big) > 0, \forall \eta, E, L$.*

The convergence rate in Theorem 5.3 consists of four terms:

- The first term $\mathcal{O}\left(\frac{\|w_g^{(0)} - w^*\|^2}{R\eta E}\right)$ is the initialization error term that depends on the total communication rounds $R$ and local learning epoch $E$. This term is fixed among all FL algorithms and independent of the FedLAG hyperparameters.

- The second term $\mathcal{O}\left(\eta\sigma_*^2\right)$ is the noise at optimum. This term reveals that the prediction at the optimum always make a certain variance.

- The third term $\mathcal{O}\left(\eta^2 E(E-1)L\sigma_*^2\right)$ refers to the user drift error, which is the error induced by the divergence when the user drift toward their specific domain characteristics. It affects by the data characteristics and FL hyperparameters such as the number of epoch $E$ and learning rate $\eta$.

- The last term $\mathcal{O}\left(A\sum_{u=1}^{U}\sum_{v=1}^{U}\sum_{l\in\mathbb{L}_p}\|h_{p,u}^{(r)}\|(\|h_{p,u}^{(r)}\| - \cos\Phi_{u,v}^{(l)}\|h_{p,v}^{(r)}\|)\right)$ is the personalized loss improvement term. This loss shows the improvement in loss thanks to the disentanglement in FL models into GAP and PL. We can see that this term consistently reduces the bound and thus creates an absolute improvement to the FL system regardless of whether the FL algorithm is integrated into it.

From the Theorem 5.3, we can have the following remarks:

**Remark 5.4.** When the conflict does not occur (e.g., the data is IID), no layer being assigned to the PL subset, which makes $\sum_{l\in\mathbb{L}_p}\|h_{p,u}^{(r)}\|(\|h_{p,u}^{(r)}\| - \cos\Phi_{u,v}^{(l)}\|h_{p,v}^{(r)}\|)$ becomes 0. Therefore, the convergence of the FedLAG reduces to $\mathcal{O}\left(\frac{3\|w_g^{(0)} - w^*\|^2}{R\eta E}\right) + \mathcal{O}\left(\eta\sigma_*^2\right) + \mathcal{O}\left(\eta^2 E(E-1)L\sigma_*^2\right)$.

**Remark 5.5.** When we set the top $k$ layers to 0, the sum of $\sum_{u=1}^{U}\sum_{v=1}^{U}\sum_{l\in\mathbb{L}_p}\|h_{p,u}^{(r)}\|(\|h_{p,u}^{(r)}\| - \cos\Phi_{u,v}^{(l)}\|h_{p,v}^{(r)}\|)$ becomes 0, therefore, the convergence of FedLAG reduces to $\mathcal{O}\left(\frac{3\|w_g^{(0)} - w^*\|^2}{R\eta E}\right) + \mathcal{O}\left(\eta\sigma_*^2\right) + \mathcal{O}\left(\eta^2 E(E-1)L\sigma_*^2\right)$.

**Remark 5.6.** From the equation, as the number of layers becomes large (i.e., over-parameterization), the right hand side may close to 0. However, in Appendix H.6, we prove that the value of the last term is agnostic to the number of parameters in the network model. The reason is because as the number of layers increase, the layer-wise gradient norm will decrease respectively.

**Remark 5.7.** Increasing $E$ and $\eta$ amplifies the value of $A$, boosting the improvement term. However, this enhancement affects local learning in distributed users. For instance, a high value of $E$ introduces bias towards local characteristics, causing users to forget global knowledge learned through aggregation, which loses the generality of the pFL concept.

**Remark 5.8.** The FedLAG's main contribution is on the server. Therefore, FedLAG can be integrated with any other FL algorithms to improve other algorithms' performance. The theoretical results of integration between FedLAG and other FL algorithms are demonstrated in Appendix D.

# 6 EXPERIMENT SETUP

## 6.1 DATASETS

To conduct a fair comparison among methods, we consider four different context data sets including MNIST ($28 \times 28$, 10 modalities) (LeCun et al., 1998), CIFAR10 ($32 \times 32 \times 3$, 10 modalities) (Krizhevsky, 2012), EMNIST ($28 \times 28$, 62 modalities) (Cohen et al., 2017), and CIFAR100 ($32 \times 32 \times 3$, 100 modalities) (Krizhevsky, 2012). Otherwise, each data set is divided into several parts corresponding to the number of users by randomly distinguished distribution. Each divided part is identical to the others to ensure that non-IID term of FL problem. Each user owns each part containing a train and test set, to which no data augmentation method is applied.

Table 1: Test accuracy comparison among baselines and our proposed method on 4 datasets (i.e., MNIST, CIFAR10/100, EMNIST). The data set splitting method is selected from the Dirichlet sampling with replacement. The experimental setups are 100 users at 20% user participation (i.e. $\alpha \in \{0.1, 0.5\}$). Each result is averaged after 5 times.

| Setting | non-IID ($\alpha = 0.1$) | | | | non-IID ($\alpha = 0.5$) | | | |
|---|---|---|---|---|---|---|---|---|
| Problem Method | MNIST | CIFAR10 | CIFAR100 | EMNIST | MNIST | CIFAR10 | CIFAR100 | EMNIST |
| FedLAG (Ours) | **97.18 ± 0.13** | **85.21 ± 0.29** | **51.35 ± 0.02** | **96.34 ± 0.25** | **96.68 ± 0.01** | **69.26 ± 0.09** | **33.75 ± 0.14** | **91.42 ± 0.18** |
| PerAvg | 89.11 ± 0.17 | 81.25 ± 0.20 | 43.21 ± 0.28 | 84.84 ± 0.20 | 94.89 ± 0.06 | 61.25 ± 0.14 | 23.98 ± 0.25 | 89.70 ± 0.14 |
| FedROD | 97.02 ± 0.29 | 81.72 ± 0.16 | 46.17 ± 0.06 | 96.02 ± 0.08 | 94.87 ± 0.12 | 68.47 ± 0.19 | 26.76 ± 0.08 | 90.70 ± 0.28 |
| FedPAC | 89.12 ± 0.04 | 83.13 ± 0.01 | 44.77 ± 0.06 | 89.63 ± 0.04 | 94.60 ± 0.18 | 63.01 ± 0.04 | 25.42 ± 0.25 | 88.44 ± 0.61 |
| FedBABU | 95.92 ± 0.26 | 80.75 ± 0.08 | 42.59 ± 0.03 | 84.63 ± 0.19 | 91.42 ± 0.03 | 65.12 ± 0.18 | 21.54 ± 0.30 | 87.35 ± 0.10 |
| FedAvg | 90.61 ± 0.11 | 65.47 ± 0.17 | 41.28 ± 0.09 | 85.02 ± 0.30 | 88.43 ± 0.07 | 59.01 ± 0.26 | 15.99 ± 0.10 | 86.35 ± 0.16 |
| FedCAC | 96.77 ± 0.03 | 84.62 ± 0.19 | 47.22 ± 1.52 | 96.01 ± 0.15 | 96.33 ± 0.11 | 68.93 ± 0.19 | 26.12 ± 0.04 | 91.10 ± 0.18 |
| FedDBE | 95.73 ± 0.11 | 83.76 ± 0.09 | 50.12 ± 0.12 | 95.04 ± 0.05 | 95.52 ± 0.11 | 67.75 ± 0.04 | 25.43 ± 0.04 | 90.14 ± 0.08 |
| GPFL | 94.15 ± 0.10 | 82.11 ± 0.25 | 49.85 ± 0.01 | 93.25 ± 0.21 | 93.56 ± 0.08 | 66.38 ± 0.07 | 24.01 ± 0.11 | 88.45 ± 0.15 |
| FedAS | 95.36 ± 0.12 | 84.11 ± 0.02 | 50.26 ± 0.03 | 93.25 ± 0.19 | 94.28 ± 0.18 | 67.38 ± 0.14 | 32.01 ± 0.04 | 89.45 ± 0.32 |
| FedAF | 95.24 ± 0.15 | 84.35 ± 0.03 | 50.11 ± 0.04 | 93.10 ± 0.20 | 94.40 ± 0.17 | 67.55 ± 0.13 | 32.07 ± 0.05 | 89.50 ± 0.30 |

Table 2: Accuracy of FedLAG under different settings of top $k$ layers. The evaluations are conducted with $U = 100$.

| Setting Dataset | k = 0 | k = 3 | k = 5 | k = 10 | k = 15 |
|---|---|---|---|---|---|
| Cifar10 | 45.47 | 83.62 | 85.21 | **85.46** | 84.77 |
| Cifar100 | 51.24 | 65.91 | **67.52** | 67.37 | 66.82 |

Table 3: Accuracy of FedLAG under different settings of hyper-parameter $\xi$. The evaluations are conducted with $U = 100$.

| Setting Dataset | $\xi = 0$ | $\xi = -0.1$ | $\xi = -0.2$ | $\xi = -0.3$ |
|---|---|---|---|---|
| Cifar10 | 85.05 | **85.62** | 84.18 | 83.26 |
| Cifar100 | 64.13 | **65.11** | 64.55 | 63.37 |

## 6.2 BASELINES

To assess the robustness of our proposed FedLAG, we run evaluations on 5 other baselines (i.e., FedAvg (McMahan et al., 2017), PerAvg (Fallah et al., 2020), FedBABU (Oh et al., 2022), FedPAC (Xu et al., 2023), and FedRoD (Chen & Chao, 2022), FedCAC (Wu et al., 2023), FedDBE (Zhang et al., 2023b), GBFL (Zhang et al., 2023c), which are all trained from scratch using ResNet18 (He et al., 2016). We apply the same settings on all baselines to achieve the fairest experimental evaluations. In detail, each method is conducted in with unbalanced data distribution and non-IID scenarios, along with different sampling rates $\alpha = 0.1\%$, and $0.5\%$. Otherwise, various numbers of users are applied to perform a fair comparison among methods (i.e. $20, 40, 60, 80, 100$), corresponding with the different number of global rounds: $100, 200, 400, 600$, and $800$ rounds, respectively.

## 7 EXPERIMENTAL EVALUATIONS

### 7.1 OVERALL PERFORMANCE

To evaluate the overall performance of the FL system, we compute average results across users. Table 1 provides a detailed overview of comprehensive performance metrics, where $\alpha$ represents the Dirichlet coefficient. The table explores the FL performance of two settings: 1) varying participation ratio and 2) different levels of heterogeneity.

### 7.1.1 DIFFERENT HETEROGENEITY LEVELS

We assess at two heterogeneity levels, i.e., $\alpha = 0.1, 1$. The table reveals that FedLAG outperforms other baselines, with improvements ranging from an average of $5 - 7\%$ to $15\%$ when $\alpha = 0.5$. The resilience of FedLAG becomes more evident in a more challenging setting, namely $\alpha = 0.1$, where it demonstrates a substantial improvement over other baselines, averaging $10\%$ to $40\%$. Among the competing baselines, FedRoD poses a notable challenge. This is because, in FedRoD, the authors employ disentanglement in personalized and generic models across different data settings (same as our work). In contrast, our research incorporates adaptive control over the two sub-models, resulting in superior performance compared to FedRoD.

### 7.1.2 DIFFERENT USER PARTICIPATION RATIO

We evaluated under five different participation ratio, i.e., $\{20\%, 40\%, 60\%, 80\%, 100\%\}$, and illustrated as in Table 7. As it can easily be seen from the table, our proposed FedLAG can achieve significantly higher performance, as opposed to other baselines (i.e., an average of $20\%$ up to $40\%$ in performance improvement). The improvement is more significant when we employ the algorithm in more challenging data sets, which showcases a more divergence in data characteristics among users. The detailed comparisons between FedLAG and other baseline models in terms of comprehensive training are presented in Appendix E.

## 7.2 ABLATION TEST

### 7.2.1 INTEGRATABILITY

In this section, we evaluavate FedLAG 's compatibility into other FL baselines on the Cifar10 dataset, and illustrate the overall performance as in Figure 5. The integration of FedLAG showcase a significantly better accuracy over baselines.

| Settings | Cifar-10 | | Cifar-100 | |
|---|---|---|---|---|
| | Acc. ($\uparrow$) | PD ($\downarrow$) | Acc. ($\uparrow$) | PD ($\downarrow$) |
| First-2 | $72.81 \pm 0.33$ | 12.40 | $17.35 \pm 0.04$ | 20.00 |
| First-4 | $71.15 \pm 0.16$ | 14.06 | $19.21 \pm 0.05$ | 18.14 |
| Last-2 | $82.47 \pm 0.18$ | 2.74 | $27.38 \pm 0.11$ | 9.97 |
| Last-4 | $80.75 \pm 0.25$ | 4.46 | $22.54 \pm 0.02$ | 14.81 |
| Middle-2 | $73.92 \pm 0.17$ | 11.29 | $15.43 \pm 0.07$ | 21.92 |
| Middle-4 | $70.15 \pm 0.15$ | 15.06 | $12.28 \pm 0.03$ | 25.07 |
| FedLAG | $85.21 \pm 0.29$ | 0 | $37.35 \pm 0.02$ | 0 |

Figure 4: The comparison of FedLAG with different fixed layer disentanglement. First-$K$ means we fix first $K$ layers for the disentanglement, Last-$K$ means we fix last $K$ layers, Middle-$K$ means we fix $K$ layers at the middle of the network.

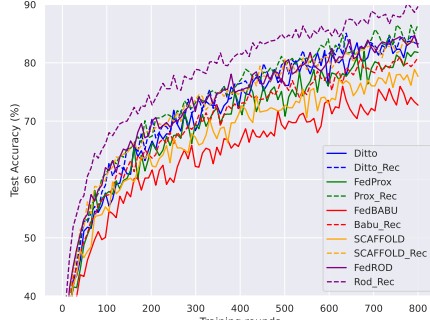

Figure 5: Convergence of FedLAG integration.

### 7.2.2 IS FEDLAG MORE EFFICIENT THAN FIXED LAYER DISENTANGLEMENT?

To prove the robustness of FedLAG over fixed layer disentanglement techniques, we conduct experiments on FedBABU, and fix $K$ layers for personalized layers. Tab. 4 demonstrates that the fixing the last $K$ layers generally yields the most efficient performance. Other settings show significant drop in performance, compared to that of the FedLAG. This observation aligns with the assumption that gradient conflicts tend to be concentrated in the final layers, though not all layers with high conflict scores are necessarily the last layers. In total, we can obviously see that the adaptive layer disentanglement of FedLAG shows a significant robustness over fixed layer disentanglement.

### 7.2.3 TOP $k$ SCORE LAYER

We conducted experiments with varying values of $k$, and the results are presented in Table 2. As shown in the table, selecting the top $k$ layers leads to significantly improved performance compared to the case where $k = 0$. This enhancement is notable because when $k = 0$, the algorithm reduces to the FedAvg algorithm, lacking the robustness inherent in the GDA algorithm. Our analysis reveals that gradient conflicts predominantly affect only a small number of layers in the FL model. Consequently, setting $k = 5, 10, 15$ does not result in a significant performance improvement, as the most conflicted gradients are already captured in the initial layers with the highest conflicts. Detailed results are presented in Appendix F.2.

### 7.2.4 CONFLICT GRADIENT SCORE

Table 3 demonstrates the performance of FedLAG under different conflict score $\xi$. As observed in the table, there is negligible performance disparity despite different gradient scores being applied. This is because the score primarily impacts the frequency of gradient conflicts across various model

layers, yet it does not alter the distribution of conflicted gradients throughout the model layers. A high value of $\xi$ (e.g., 0) results in performance drops due to an excessive inclusion of gradient pairs with small conflicts. Some of these pairs are beneficial for learning common structures and should not be excluded. Conversely, a too-small value for $S$ (e.g., $-0.3$) also leads to performance degradation by ignoring many gradient pairs with significant conflicts, which can be detrimental to the learning process. Detailed results are presented in Appendix F.3.

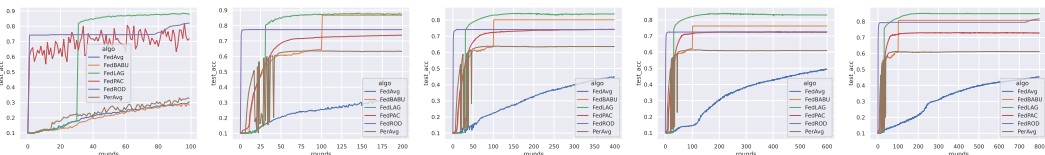

Figure 6: The figures illustrate the performance without pretrained model of FL-LAG vs. various baselines on CIFAR10. The evaluation is implemented on different numbers of users (from left to right, the number of users is $\{100\%, 80\%, 60\%, 40\%, 20\%\}$, respectively). $\alpha = 0.1$.

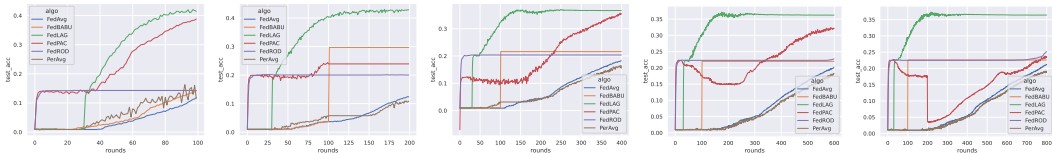

Figure 7: The figures illustrate the performance without pretrained model of FL-LAG vs. various baselines on CIFAR100. The evaluation is implemented on different numbers of users (from left to right, the number of users is $\{100\%, 80\%, 60\%, 40\%, 20\%\}$, respectively). $\alpha = 0.1$.

### 7.2.5 Efficiency when training without pretrained models

In this section, we evaluate FL without pretrained models on the CIFAR-10 and CIFAR-100 datasets to assess FL performance when training from scratch (see Figs. 6, 7). We only apply the FedLAG after first 30 rounds to sample the very first gradient trajectories for the initial process of GDA. Most baseline models encounter significant challenges when trained without pre-trained parameters. In contrast, our FedLAG consistently demonstrates superior performance, achieving the highest results alongside FedRod and FedPAC. The significant superiority of FedLAG is shown when training on the challenging CIFAR-100 dataset, which contains a large number of labels. While other federated learning baselines struggle to exceed random performance, FedLAG rapidly surpasses this threshold and achieves effective convergence in fewer than 100 communication rounds.

## 8 Conclusion

Current researches on layer disentanglement in pFL typically require extensive fine-tuning to achieve optimal separation between generic and personalized layers. In our study, we introduce an adaptive approach to disentangle these layers by leveraging a well-established principle in multi-task learning, namely, conflicting gradients. To address conflicting gradients among users without incurring communication overhead from user-to-user interactions, we propose a novel data-free gradient divergence analysis method performed on the server. This technique enhances federated learning performance by enabling the selective assignment of network layers to personalization when layer-wise gradients are in conflict, and to generic layers otherwise. Our proposed method, FedLAG, demonstrates significant improvements over current baselines in both accuracy and convergence time.

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

# A   NOTATIONS

Table 4: Abbreviations

| Symbol | Description |
|--------|-------------|
| $U$ | Number of users |
| $\eta$ | Learning rate |
| $w$ | Model parameter |
| $E$ | Number of training epochs for each user per round |
| $\xi$ | Threshold for layer-wise gradient conflict score |
| $L$ | Number of model layers |
| $R$ | Number of rounds |
| $w*$ | Optimal global model |
| $\sigma_*^2$ | Upper bound of variance of user gradients at optimum |
| $w_g^{(r)}$ | Global parameter at round $t$ |
| $w_u^{(r,e)}$ | Local parameter at round $t$, training at epoch $e$ |
| $\cos \Phi_{u,v}^{(l)}$ | Cosine of gradient of user $u$ and $v$ w.r.t the $l^{th}$ layer |
| $\theta_{s,u}^{(r)}, \theta_{s,g}^{(r)}$ | Model layers that is assigned for global aggregation |
| $\theta_{p,u}^{(r)}, \theta_{p,g}^{(r)}$ | Model layers that is assigned for personalized purpose |
| $\theta_{l,u}^{(r)}$ | $l^{th}$ Layer in model parameter |
| $h_u^{(r)}$ | gradient of user $u$ on round $r$ |
| $h_{l,u}^{(r)}$ | gradient w.r.t the $l^{th}$ layer |
| $P$ | Number of parameters of each pFL model |
| $P_l$ | Number of parameters on layer $l^{th}$ |
| $\mathbb{L}_p, \mathbb{L}_g$ | The set of coefficients in the personalized and global model |
| $k$ | Number of selected layers for personalization |
| $w_{\text{LAG},u}^{(r)}$ | Layer-wise Personalized Model of user $u$ |
| $GC_\xi(l)$ | Layer-wise gradient conflict score |

# B  RELATED WORKS

**Gradient-based Multi-task Learning.** In MTL, the AI model consists of two distinguished groups: 1) *shared encoder* that is common to all tasks, and 2) *task-specific decoders* which are designed and learned independently for each task. Addressing the contemporary challenge of task conflicts in MTL, numerous approaches have been devised to mitigate the aforementioned issue, which can be categorized in two main directions, i.e., task loss balancing and gradient manipulation. PCGrad (Yu et al., 2020a), projects each gradient onto the normal plane of another gradient and employs the average of these projected gradients for updates. GradDrop (Chen et al., 2020), randomly drops some elements of gradients based on element-wise conflicts. CAGrad (Liu et al., 2021), ensures convergence to a minimum of the average loss across tasks through gradient manipulation. RotoGrad (Javaloy & Valera, 2022), re-weights task gradients and rotates the shared feature space to mitigate conflicts. RECON (Shi et al., 2023), leverages gradient information to modify network structure and address task conflicts at their core.

**Mitigating non-IID in Federated Learning.** Prior work has investigated how to improve the FL robustness against non-IID data. FedPAC (Xu et al., 2023) training personalized models by exploiting a better feature extractor and user-specific classifier collaboration. FedProx (Li et al., 2020) adds a proximal term to the local training objective to keep updated parameters close to the original downloaded model. SCAFFOLD (Karimireddy et al., 2020) introduces control variates to correct the drift in local updates. MOON (Li et al., 2021) adopts contrastive loss to improve representation learning.

**Layer disentanglement in Federated Learning.** Oh et al. (2022) decompose the entire local network into the body (extractor), which is related to universality, and the head (classifier), which is related to personalization. Collins et al. (2021) proposes a method where the entire network is trained sequentially during local updates, but only the body is aggregated. During the local update phase, each client first trains the head using the aggregated representation. Then, within the same epoch, the client trains the body using its own head. FedRoD (Chen & Chao, 2022) proposes to use the balanced softmax for learning generic models and vanilla softmax for personalized heads. Xu et al. (2023) design an objective function to constraint the body with the task of learning invariant features. However, most layer disentanglement approaches demand significant effort to identify the optimal layer selection, often leading to arbitrary choices for the body and head layers. Currently, there is no clear method for determining which layers should be personalized and which should remain generic.

**Mitigating negative transfer in Federated Learning.** Recently, negative transfer has been discovered to be one of the most crucial issue in FL. FedCollab (Bao et al., 2023) minimizes the pair-wise distribution distances between users to alleviate the negative transfer among clients. DisentAFL (Chen & Zhang, 2024) integrates mixtures of experts to selectively aggregate clients' representations. To this end, the FL system can aggregate the fine-grained inter-client relationships to achieve sufficient positive transfer while avoiding negative transfer. However, we believe that current methods lack a strong theoretical foundation that directly addresses negative transfer among clients.

## C  DETAIL OF MODELS USED

| Dataset | Model | 1 Initial Conv Output Channel | Residual Blocks Blocks/ Group | Residual Blocks Output Channels/ Group | 3 FC Layers Output Dimension | Activation Function | Total Model Size (MB) |
|---|---|---|---|---|---|---|---|
| MNIST | Resnet-9 | 32 | (1,2,2,1) | (32,64,128,256) | (256,128,10) | ReLU | 6.81 |
| EMNIST | Resnet-9 | 32 | (1,2,2,1) | (32,64,128,256) | (256,128,10) | ReLU | 6.81 |
| CIFAR-10 | Resnet-20 | 64 | (3,3,3) | (64,128,256) | (256,128,10) | ReLU | 17.54 |
| CIFAR-100 | Resnet-20 | 64 | (2,3,4) | (64,128,256) | (256,128,100) | ReLU | 17.54 |

Table 5: Model employed for MNIST, EMNIST, CIFAR-10, CIFAR-100 Datasets.

## D  EXTENSIVE THEORETICAL RESULTS

As current FL concepts are mostly gradient-based optimization approaches, the convergence proofs of all FL approaches are proved in a similar approach. Therefore, for simplicity, we do not prove all the theoretical results comprehensively. As being proved in Appendix H.4, our LAG always give the improvement of $-\frac{1}{U}\sum_{u=1}^{U}\sum_{l\in\mathbb{L}_p}\|h_{p,u}^{(r)}\|(\|h_{p,u}^{(r)}\| - \cos\Phi_{u,v}^{(l)}\|h_{p,v}^{(r)}\|)$ on any global loss function. Therefore, by apply this theorem to the proof in other approaches (e.g., pFedMe (T. Dinh et al., 2020), FedEXP (Jhunjhunwala et al., 2023), SCAFFOLD (Karimireddy et al., 2020)), and apply the model disentanglement to apply the loss improvement lemma, we can have the results as following table:

| Method | Convergence Rate |
|---|---|
| FedAvg (McMahan et al., 2017) | $\mathcal{O}\left(\frac{3\|w_g^{(0)}-w^*\|^2}{R\eta E}\right) + \mathcal{O}\left(\eta\sigma_*^2\right) + \mathcal{O}\left(\eta^2 E(E-1)L\sigma_*^2\right)$ |
| FedAvg + LAG (Ours) | $\mathcal{O}\left(\frac{3\|w_g^{(0)}-w^*\|^2}{R\eta E}\right) + \mathcal{O}\left(\eta\sigma_*^2\right) + \mathcal{O}\left(\eta^2 E(E-1)L\sigma_*^2\right)$ $-\mathcal{O}\left(\|h_{p,u}^{(r)}\|(\|h_{p,u}^{(r)}\| - \cos\Phi_{u,v}^{(l)}\|h_{p,v}^{(r)}\|)\right)$ |
| FedEXP (Jhunjhunwala et al., 2023) | $\mathcal{O}\left(\frac{3\|w_g^{(0)}-w^*\|^2}{\eta_l E\sum_{r=0}^{R-1}\eta_g^{(r)}}\right) + \mathcal{O}\left(\eta_l\sigma_*^2\right) + \mathcal{O}\left(\eta_l^2 E(E-1)L\sigma_*^2\right)$ |
| FedEXP + LAG (Ours) | $\mathcal{O}\left(\frac{3\|w_g^{(0)}-w^*\|^2}{\eta_l E\sum_{r=0}^{R-1}\eta_g^{(r)}}\right) + \mathcal{O}\left(\eta_l\sigma_*^2\right) + \mathcal{O}\left(\eta_l^2 E(E-1)L\sigma_*^2\right)$ $-\mathcal{O}\left(\|h_{p,u}^{(r)}\|(\|h_{p,u}^{(r)}\| - \cos\Phi_{u,v}^{(l)}\|h_{p,v}^{(r)}\|)\right)$ |
| pFedMe (T. Dinh et al., 2020) | $\mathcal{O}\left(3\|w_g^{(0)} - w^*\|^2\mu_F e^{-\hat{\eta}_1\mu_F T/2}\right) + \mathcal{O}\left(\frac{(N/S-1)\sigma_{F,1}^2}{\mu_F TN}\right)$ $+\mathcal{O}\left(\frac{(R\sigma_{F,1}^2+\delta^2\lambda^2)\kappa_F}{R(T\beta\mu_F)^2}\right) + \mathcal{O}(\frac{\lambda^2\delta^2}{\mu_F})$ |
| pFedMe + LAG (Ours) | $\mathcal{O}\left(3\|w_g^{(0)} - w^*\|^2\mu_F e^{-\hat{\eta}_1\mu_F T/2}\right) + \mathcal{O}\left(\frac{(N/S-1)\sigma_{F,1}^2}{\mu_F TN}\right)$ $+\mathcal{O}\left(\frac{(R\sigma_{F,1}^2+\delta^2\lambda^2)\kappa_F}{R(T\beta\mu_F)^2}\right) + \mathcal{O}(\frac{\lambda^2\delta^2}{\mu_F})$ $-\mathcal{O}\left(\|h_{p,u}^{(r)}\|(\|h_{p,u}^{(r)}\| - \cos\Phi_{u,v}^{(l)}\|h_{p,v}^{(r)}\|)\right)$ |
| SCAFFOLD (Karimireddy et al., 2020) | $\mathcal{O}(\frac{\sigma}{\mu SK_\epsilon}) + \mathcal{O}(\frac{1}{\mu}) + \mathcal{O}(\frac{N}{S})$ |
| SCAFFOLD + LAG (Ours) | $\mathcal{O}(\frac{\sigma}{\mu SK_\epsilon}) + \mathcal{O}(\frac{1}{\mu}) + \mathcal{O}(\frac{N}{S})$ $-\mathcal{O}\left(\|h_{p,u}^{(r)}\|(\|h_{p,u}^{(r)}\| - \cos\Phi_{u,v}^{(l)}\|h_{p,v}^{(r)}\|)\right)$ |

Table 6: Table of convergence of baselines vs. baselines + LAG.

# E DETAILED RESULTS

## E.1 FULL TABLE OF OVERALL PERFORMANCE

| Problem | **non-IID** ($\alpha = 0.1$) | | | | **non-IID** ($\alpha = 0.5$) | | | |
|---|---|---|---|---|---|---|---|---|
| Problem Method | MNIST | CIFAR10 | CIFAR100 | EMNIST | MNIST | CIFAR10 | CIFAR100 | EMNIST |
| **Participation ratio = 100%** | | | | | | | | |
| **FedLAG (Ours)** | **97.97 ± 0.17** | **88.54 ± 0.1** | **57.08 ± 0.28** | **96.73 ± 0.15** | **99.57 ± 0.10** | **87.18 ± 0.03** | **41.81 ± 0.18** | **97.57 ± 0.27** |
| PerAvg | 87.05 ± 0.13 | 83.12 ± 0.26 | 51.27 ± 0.05 | 83.34 ± 0.24 | 93.36 ± 0.05 | 83.50 ± 0.08 | 34.13 ± 0.27 | 95.74 ± 0.12 |
| FedROD | 97.74 ± 0.11 | 82.09 ± 0.04 | 49.33 ± 0.20 | 95.25 ± 0.20 | 94.88 ± 0.29 | 81.94 ± 0.10 | 38.34 ± 0.30 | 96.46 ± 0.20 |
| FedPAC | 96.64 ± 0.21 | 81.75 ± 0.17 | 53.83 ± 0.11 | 92.22 ± 0.09 | 94.36 ± 0.19 | 80.30 ± 0.13 | 38.51 ± 0.04 | 93.44 ± 0.05 |
| FedBABU | 85.46 ± 0.15 | 79.55 ± 0.27 | 51.78 ± 0.12 | 83.77 ± 0.27 | 90.62 ± 0.03 | 86.60 ± 0.15 | 33.18 ± 0.16 | 93.48 ± 0.16 |
| FedAvg | 85.77 ± 0.19 | 60.45 ± 0.02 | 47.25 ± 0.21 | 83.24 ± 0.10 | 91.90 ± 0.11 | 78.15 ± 0.05 | 32.64 ± 0.02 | 93.83 ± 0.26 |
| FedCAC | 95.90 ± 0.15 | 86.50 ± 0.08 | 53.46 ± 0.25 | 94.65 ± 0.13 | 93.12 ± 0.08 | 85.73 ± 0.02 | 39.14 ± 0.12 | 95.50 ± 0.25 |
| FedDBE | 94.90 ± 0.12 | 85.50 ± 0.06 | 55.13 ± 0.22 | 93.65 ± 0.10 | 92.26 ± 0.06 | 84.26 ± 0.01 | 38.75 ± 0.26 | 93.50 ± 0.22 |
| GPFL | 93.90 ± 0.10 | 84.50 ± 0.04 | 54.11 ± 0.20 | 92.65 ± 0.08 | 91.37 ± 0.04 | 83.12 ± 0.01 | 38.23 ± 0.10 | 95.50 ± 0.17 |
| FedAS | 97.22 ± 0.02 | 87.73 ± 0.23 | 54.92 ± 0.06 | 97.77 ± 0.01 | 98.14 ± 0.18 | 85.92 ± 0.26 | 38.19 ± 0.05 | 96.89 ± 0.22 |
| FedAF | 97.25 ± 0.04 | 86.25 ± 0.01 | 51.80 ± 0.02 | 95.25 ± 0.11 | 97.00 ± 0.02 | 85.10 ± 0.03 | 38.20 ± 0.16 | 95.50 ± 0.23 |
| **Participation ratio = 80%** | | | | | | | | |
| **FedLAG (Ours)** | **97.95 ± 0.20** | **88.02 ± 0.24** | **55.98 ± 0.10** | **95.72 ± 0.16** | **99.28 ± 0.21** | **85.25 ± 0.12** | **39.91 ± 0.09** | **96.14 ± 0.29** |
| PerAvg | 89.44 ± 0.24 | 83.61 ± 0.06 | 48.93 ± 0.04 | 83.88 ± 0.09 | 93.18 ± 0.05 | 81.59 ± 0.16 | 35.37 ± 0.23 | 94.53 ± 0.22 |
| FedROD | 97.61 ± 0.24 | 87.42 ± 0.11 | 48.18 ± 0.27 | 95.51 ± 0.09 | 94.85 ± 0.27 | 82.03 ± 0.18 | 35.26 ± 0.24 | 97.92 ± 0.28 |
| FedPAC | 95.05 ± 0.23 | 83.81 ± 0.11 | 50.37 ± 0.21 | 93.92 ± 0.07 | 94.24 ± 0.15 | 80.87 ± 0.19 | 36.12 ± 0.14 | 93.55 ± 0.10 |
| FedBABU | 96.51 ± 0.08 | 86.7 ± 0.28 | 50.69 ± 0.29 | 83.58 ± 0.18 | 90.03 ± 0.25 | 83.52 ± 0.29 | 37.24 ± 0.03 | 92.53 ± 0.12 |
| FedAvg | 89.88 ± 0.06 | 63.36 ± 0.18 | 46.46 ± 0.05 | 83.71 ± 0.15 | 92.18 ± 0.01 | 75.75 ± 0.21 | 30.13 ± 0.26 | 92.94 ± 0.19 |
| FedCAC | 95.82 ± 0.14 | 86.47 ± 0.09 | 52.92 ± 0.27 | 94.60 ± 0.12 | 92.92 ± 0.07 | 85.04 ± 0.03 | 37.84 ± 0.14 | 94.46 ± 0.23 |
| FedDBE | 94.87 ± 0.13 | 85.44 ± 0.07 | 54.94 ± 0.23 | 93.58 ± 0.09 | 91.97 ± 0.05 | 84.04 ± 0.02 | 37.62 ± 0.13 | 93.44 ± 0.21 |
| GPFL | 93.84 ± 0.09 | 84.44 ± 0.05 | 53.93 ± 0.19 | 92.58 ± 0.07 | 90.95 ± 0.03 | 83.07 ± 0.14 | 38.71 ± 0.11 | 92.46 ± 0.18 |
| FedAS | 96.62 ± 0.07 | 86.23 ± 0.25 | 53.73 ± 0.01 | 95.77 ± 0.25 | 97.28 ± 0.08 | 84.92 ± 0.07 | 37.89 ± 0.11 | 93.45 ± 0.15 |
| FedAF | 97.25 ± 0.04 | 86.25 ± 0.01 | 51.80 ± 0.02 | 95.25 ± 0.11 | 96.00 ± 0.09 | 79.10 ± 0.07 | 37.20 ± 0.02 | 91.50 ± 0.20 |
| **Participation ratio = 60%** | | | | | | | | |
| **FedLAG (Ours)** | **97.39 ± 0.20** | **87.25 ± 0.15** | **55.03 ± 0.22** | **96.88 ± 0.14** | **98.87 ± 0.26** | **83.24 ± 0.26** | **37.98 ± 0.24** | **94.82 ± 0.26** |
| PerAvg | 89.04 ± 0.08 | 83.85 ± 0.28 | 46.52 ± 0.09 | 84.18 ± 0.04 | 94.52 ± 0.22 | 79.76 ± 0.10 | 33.09 ± 0.07 | 93.78 ± 0.09 |
| FedROD | 97.15 ± 0.19 | 83.15 ± 0.21 | 48.39 ± 0.09 | 96.5 ± 0.16 | 94.82 ± 0.30 | 79.52 ± 0.07 | 35.19 ± 0.18 | 93.46 ± 0.19 |
| FedPAC | 84.9 ± 0.04 | 83.35 ± 0.19 | 49.75 ± 0.06 | 94.74 ± 0.04 | 94.62 ± 0.17 | 76.17 ± 0.19 | 34.95 ± 0.20 | 93.10 ± 0.12 |
| FedBABU | 95.85 ± 0.07 | 80.17 ± 0.15 | 49.61 ± 0.15 | 83.76 ± 0.02 | 90.78 ± 0.07 | 74.91 ± 0.27 | 31.96 ± 0.06 | 92.11 ± 0.12 |
| FedAvg | 90.04 ± 0.14 | 64.9 ± 0.05 | 46.51 ± 0.22 | 84.23 ± 0.05 | 93.47 ± 0.20 | 68.12 ± 0.07 | 22.24 ± 0.03 | 92.45 ± 0.18 |
| FedCAC | 95.75 ± 0.16 | 86.43 ± 0.10 | 51.98 ± 0.45 | 94.55 ± 0.14 | 92.85 ± 0.09 | 75.14 ± 0.03 | 35.20 ± 0.16 | 93.40 ± 0.27 |
| FedDBE | 94.80 ± 0.14 | 85.40 ± 0.07 | 54.23 ± 0.24 | 93.50 ± 0.11 | 91.90 ± 0.07 | 76.42 ± 0.02 | 34.81 ± 0.14 | 92.40 ± 0.23 |
| GPFL | 93.80 ± 0.11 | 84.40 ± 0.06 | 53.85 ± 0.21 | 92.50 ± 0.09 | 90.90 ± 0.05 | 76.95 ± 0.03 | 34.65 ± 0.12 | 91.40 ± 0.19 |
| FedAS | 96.24 ± 0.11 | 85.79 ± 0.12 | 51.24 ± 0.23 | 95.34 ± 0.16 | 96.28 ± 0.22 | 76.92 ± 0.01 | 32.89 ± 0.09 | 91.26 ± 0.05 |
| FedAF | 96.25 ± 0.07 | 85.25 ± 0.01 | 51.00 ± 0.02 | 94.25 ± 0.15 | 95.20 ± 0.12 | 78.30 ± 0.09 | 32.50 ± 0.02 | 90.50 ± 0.24 |
| **Participation ratio = 40%** | | | | | | | | |
| **FedLAG (Ours)** | **97.55 ± 0.15** | **86.11 ± 0.62** | **53.19 ± 0.09** | **96.14 ± 0.07** | **96.17 ± 0.22** | **76.34 ± 0.08** | **36.39 ± 0.16** | **93.58 ± 0.13** |
| PerAvg | 89.43 ± 0.02 | 81.44 ± 0.01 | 44.38 ± 0.23 | 84.17 ± 0.02 | 91.71 ± 0.12 | 73.57 ± 0.07 | 32.10 ± 0.27 | 90.09 ± 0.29 |
| FedROD | 97.15 ± 0.07 | 82.53 ± 0.19 | 47.76 ± 0.42 | 95.73 ± 0.18 | 94.72 ± 0.03 | 78.03 ± 0.07 | 35.21 ± 0.16 | 93.30 ± 0.16 |
| FedPAC | 84.24 ± 0.29 | 82.63 ± 0.21 | 48.38 ± 0.13 | 86.42 ± 0.19 | 93.57 ± 0.07 | 75.91 ± 0.27 | 32.26 ± 0.28 | 93.27 ± 0.02 |
| FedBABU | 95.34 ± 0.09 | 76.19 ± 0.06 | 45.01 ± 0.04 | 83.97 ± 0.07 | 91.17 ± 0.25 | 72.21 ± 0.06 | 31.46 ± 0.13 | 91.44 ± 0.14 |
| FedAvg | 90.71 ± 0.14 | 69.66 ± 0.17 | 42.21 ± 0.18 | 84.45 ± 0.19 | 93.88 ± 0.28 | 65.63 ± 0.25 | 16.05 ± 0.25 | 91.05 ± 0.03 |
| FedCAC | 96.05 ± 0.18 | 77.19 ± 0.09 | 48.15 ± 0.12 | 94.80 ± 0.14 | 93.20 ± 0.09 | 75.20 ± 0.03 | 33.45 ± 0.16 | 92.80 ± 0.27 |
| FedDBE | 95.48 ± 0.13 | 75.80 ± 0.08 | 51.27 ± 0.26 | 94.25 ± 0.12 | 92.10 ± 0.07 | 74.20 ± 0.02 | 32.74 ± 0.13 | 91.70 ± 0.22 |
| GPFL | 94.36 ± 0.12 | 74.60 ± 0.05 | 50.86 ± 0.22 | 93.17 ± 0.10 | 91.20 ± 0.06 | 73.20 ± 0.02 | 32.26 ± 0.11 | 91.52 ± 0.20 |
| FedAS | 95.76 ± 0.15 | 84.79 ± 0.22 | 50.89 ± 0.01 | 94.56 ± 0.19 | 95.28 ± 0.03 | 74.22 ± 0.17 | 32.89 ± 0.16 | 90.45 ± 0.02 |
| FedAF | 95.50 ± 0.10 | 84.50 ± 0.02 | 50.40 ± 0.02 | 93.50 ± 0.18 | 94.60 ± 0.15 | 67.70 ± 0.12 | 32.15 ± 0.04 | 89.80 ± 0.28 |
| **Participation ratio = 20%** | | | | | | | | |
| **FedLAG (Ours)** | **97.18 ± 0.13** | **85.21 ± 0.29** | **51.35 ± 0.02** | **96.34 ± 0.25** | **96.68 ± 0.01** | **69.26 ± 0.09** | **33.75 ± 0.14** | **91.42 ± 0.18** |
| PerAvg | 89.11 ± 0.17 | 81.25 ± 0.20 | 43.21 ± 0.28 | 84.84 ± 0.20 | 94.89 ± 0.06 | 61.25 ± 0.14 | 23.98 ± 0.25 | 89.70 ± 0.14 |
| FedROD | 97.02 ± 0.29 | 81.72 ± 0.16 | 46.17 ± 0.06 | 96.02 ± 0.08 | 94.87 ± 0.12 | 68.47 ± 0.19 | 26.76 ± 0.08 | 90.70 ± 0.28 |
| FedPAC | 89.12 ± 0.04 | 83.13 ± 0.01 | 44.77 ± 0.06 | 89.63 ± 0.04 | 94.60 ± 0.18 | 63.01 ± 0.04 | 25.42 ± 0.25 | 88.44 ± 0.61 |
| FedBABU | 95.92 ± 0.26 | 80.75 ± 0.08 | 42.59 ± 0.03 | 84.63 ± 0.18 | 91.42 ± 0.03 | 65.12 ± 0.18 | 21.54 ± 0.30 | 87.35 ± 0.10 |
| FedAvg | 90.61 ± 0.11 | 65.47 ± 0.17 | 41.28 ± 0.09 | 85.02 ± 0.30 | 88.43 ± 0.07 | 59.01 ± 0.26 | 15.99 ± 0.10 | 86.35 ± 0.16 |
| FedCAC | 96.77 ± 0.03 | 84.62 ± 0.19 | 47.22 ± 1.52 | 96.01 ± 0.15 | 96.33 ± 0.11 | 68.93 ± 0.19 | 26.12 ± 0.04 | 91.10 ± 0.18 |
| FedDBE | 95.73 ± 0.11 | 83.76 ± 0.09 | 50.12 ± 0.12 | 95.04 ± 0.05 | 95.52 ± 0.11 | 67.75 ± 0.04 | 25.43 ± 0.04 | 90.14 ± 0.08 |
| GPFL | 94.15 ± 0.10 | 82.11 ± 0.25 | 49.85 ± 0.01 | 93.25 ± 0.21 | 93.56 ± 0.08 | 66.38 ± 0.07 | 24.01 ± 0.11 | 88.45 ± 0.15 |
| FedAS | 95.36 ± 0.12 | 84.11 ± 0.02 | 50.26 ± 0.03 | 93.25 ± 0.19 | 94.28 ± 0.18 | 67.38 ± 0.14 | 32.01 ± 0.04 | 89.45 ± 0.32 |
| FedAF | 95.24 ± 0.15 | 84.35 ± 0.03 | 50.11 ± 0.04 | 93.10 ± 0.20 | 94.40 ± 0.17 | 67.55 ± 0.13 | 32.07 ± 0.05 | 89.50 ± 0.30 |

Table 7: Test accuracy comparison among baselines and our proposed method on 4 datasets (i.e., MNIST, CIFAR10/100, EMNIST). The data set splitting method is selected from the Dirichlet sampling with replacement (i.e. $\alpha \in \{0.1, 0.5\}$). The experimental setups are at 20%, 40%, 60%, 80%, and 100% user participation. Each result is averaged after 10 times.

# F    DETAILED ABLATION TEST

## F.1    TRAINING TIME

We have conducted a thorough evaluation of FedLAG's robustness in comparison to other baselines, focusing on convergence rate and accuracy. Additionally, we evaluated the robustness of FedLAG by computing the average computation time per round for various FL algorithms, and the results are presented in Table 8.

As depicted in the table, FedLAG demonstrates a lower computation cost compared to other baselines. This efficiency stems from our approach, which involves measuring the gradient and applying a straightforward analysis on the server. It is noteworthy that servers typically possess significantly more computational resources than individual devices. As a result, our proposed FedLAG effectively leverages the server's computation capacity, leading to substantial time savings.

The efficiency gains achieved by FedLAG make it particularly well-suited for Internet of Things (IoT) systems with constrained computational resources. In contrast, FedPAC tends to be less suitable for such environments, as it demands substantial computation time on distributed devices. Our findings emphasize the practical advantages of FedLAG in scenarios where computational efficiency is crucial, such as IoT systems with limited resources.

| Method Dataset | FedBABU | PerAvg | FedROD | FedPAC | FedLAG (Ours) |
|---|---|---|---|---|---|
| **Mnist** | $11 \pm 0.11$ | $9 \pm 0.13$ | $12 \pm 0.32$ | $342 \pm 0.06$ | $\mathbf{8 \pm 0.37}$ |
| **Cifar10** | $23 \pm 0.25$ | $19 \pm 0.62$ | $22 \pm 0.18$ | $474 \pm 0.26$ | $\mathbf{15 \pm 0.77}$ |
| **Cifar100** | $22 \pm 0.13$ | $20 \pm 0.11$ | $21 \pm 0.55$ | $698 \pm 0.77$ | $\mathbf{17 \pm 0.34}$ |
| **EMNIST** | $117 \pm 0.25$ | $\mathbf{108 \pm 0.20}$ | $114 \pm 0.70$ | $503 \pm 0.43$ | $112 \pm 0.43$ |

Table 8: Training time (second/round) of FedLAG vs. baselines.

## F.2 TOP K LAYERS

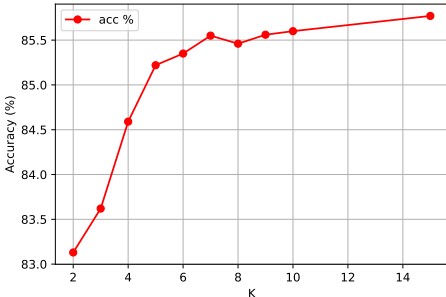
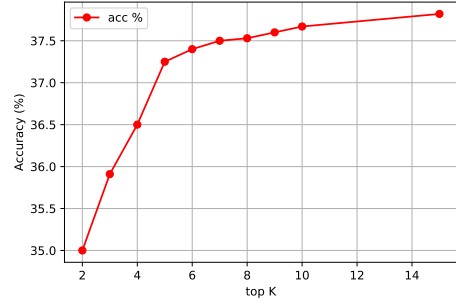

Figure 8: The impact of the top $k$ conflict layer hyperparameter on accuracy on cifar 10.

Figure 9: The impact of the top $k$ conflict layer hyperparameter on accuracy on cifar 100.

## F.3 GRADIENT SCORE

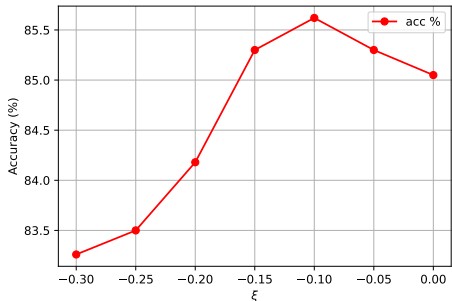
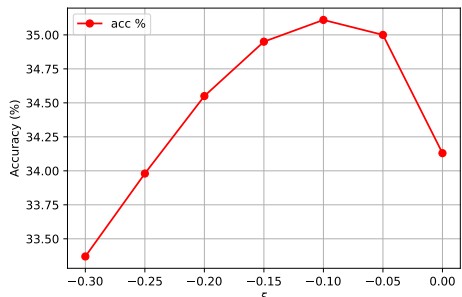

Figure 10: The impact of the conflict hyperparameter $\xi$ on accuracy on cifar 10.

Figure 11: The impact of the conflict hyperparameter $\xi$ on accuracy on cifar 100.

### F.4 CONFLICTED SCORE DURING THE TRAINING OF FEDAVG

### F.4.1 MNIST DATASET

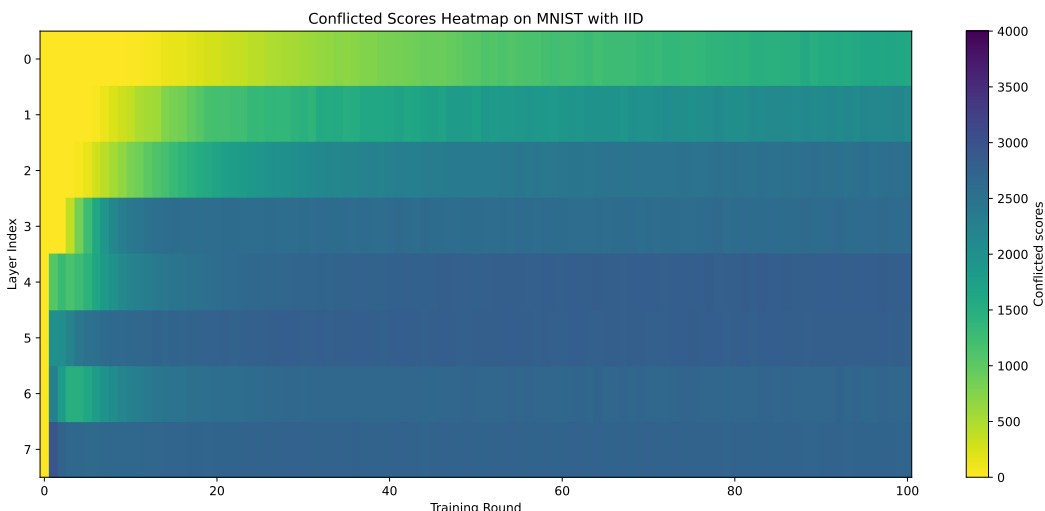

Figure 12: The GC score on each layer (Dataset MNIST - IID).

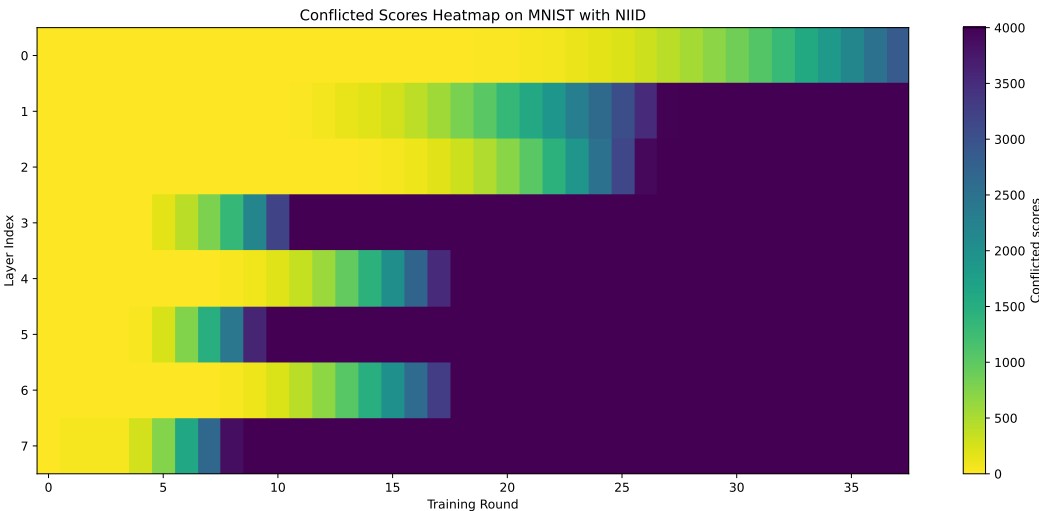

Figure 13: The GC score on each layer (Dataset MNIST - NIID $\alpha = 0.1$).

In the MNIST dataset under the IID setting shown in Fig. 12, it's clear that the gradient conflict score is close to 0 in the early rounds. This is due to training on IID settings and the simplicity of the MNIST dataset. Consequently, there's minimal negative transfer of gradient conflict among users. When compared with that of the conflicted scores in MNIST dataset with non-IID settings, we can see that the problem of conflicted scores become significant.

### F.4.2 CIFAR10 DATASET

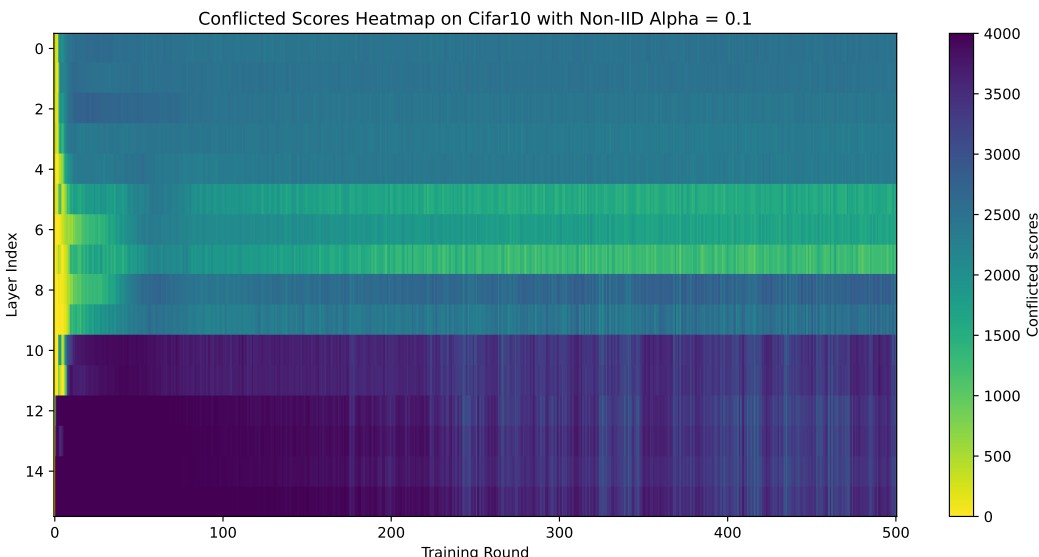

Figure 14: The GC score on each layer (Dataset Cifar10 - NonIID with $\alpha = 0.1$).

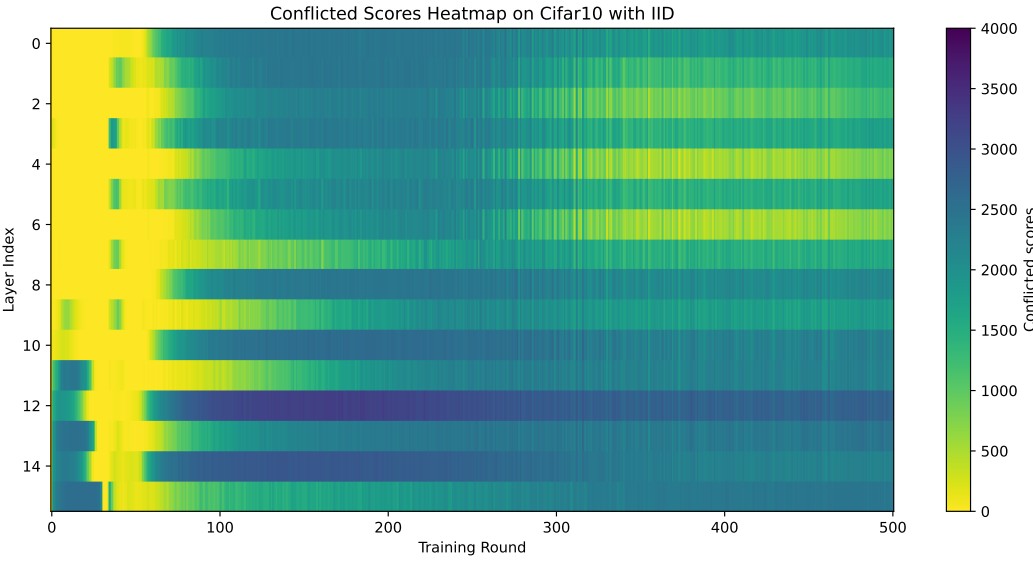

Figure 15: The GC score on each layer (Dataset Cifar10 - IID).

Fig. 14 shows the fluctuating scores observed during training on the Cifar10 dataset under non-IID conditions ($\alpha = 0.1$). We excluded layers without trainable parameters, such as Batch Normalization layers, from visualization. Initially, from round 0 to 150, the conflicting gradients are prominent in the later layers (i.e., layers 12 to 16). However, as training progresses, the intensity of conflicts in these layers diminishes, resulting in a more balanced distribution of conflicting gradients across layers 8 to 16. This indicates that the nature of conflicting scores varies throughout the training stages of FL. In the IID settings (Fig. 15), the conflicted scores is much reduced as the divergence among domains are small. However, we can easily see that the layers with high gradient conflicts are not densely distributed at the very first and the very last layers.

# G    TOY DATASET DESCRIPTION

We visualize the training data in 16, and 17. In the FDG setting, the users are from different domains. To this end, we design the data where the point are distributed into rectangular with different size and shape. The rationale of designing the data distribution is as follows:

- The global dataset consists of two classes from two rectangular, which has the classification boundary is equal to $y = 0$.

- Each domain-wise dataset has different classification boundary (e.g., $x = -6$ for domain 1). We add the noisy data on every domains so that the user assign to each domain will tend to learn the local boundary instead of the global boundary. Thus, we can observe the gradient divergence more clearly, as the global boundary is not the optimal solution when learn on local dataset.

- All of the local classification boundary is orthogonal from the global classification boundary, thus, we can make the learning more challenging despite the simplicity of the toy dataset.

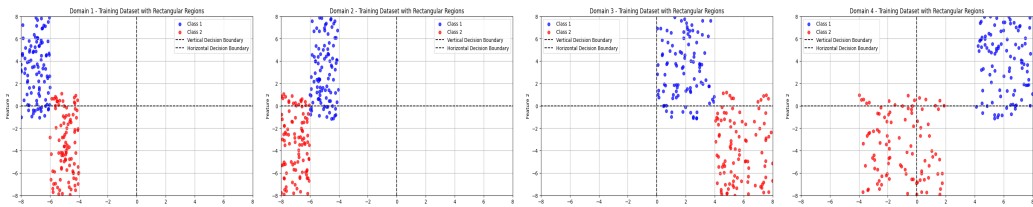

Figure 16: Illustration of users with different domains.

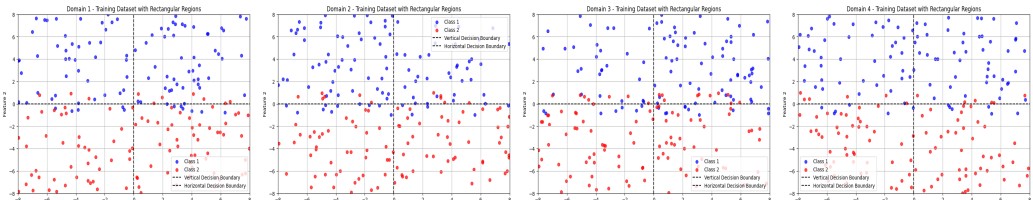

Figure 17: Illustration of users with same domains.

# H    PROOF

## H.1    ASSUMPTIONS AND DEFINITIONS

To come up with our theoretical analysis, we first adopt the following assumptions:

**Assumption H.1** ($L$-smooth). Local objective $F_u(w)$ is differentiable and $L$-smooth for all $u \in \{1, 2, \ldots, U\}$, i.e., $\|\nabla \mathcal{L}_u(w) - \nabla \mathcal{L}_u(w')\| \leq L\|w - w'\|, \ \forall w, w' \in \mathbb{R}^d$.

**Assumption H.2** ($\mu$-strongly convex). Local objective $F_u(w)$ is differentiable and $L$-smooth for all $u \in \{1, 2, \ldots, U\}$, i.e., $\|\nabla \mathcal{L}_u(w) - \nabla \mathcal{L}_u(w')\| \geq \mu\|w - w'\|, \ \forall w, w' \in \mathbb{R}^d$.

**Assumption H.3** (Bounded data heterogeneity at optimum). The norm of the user gradients at the global optima $w^*$ is bounded as follows: $\frac{1}{U}\sum_{u=1}^{U}\|\nabla \mathcal{L}_u(w^*)\|^2 \leq \sigma_*^2$.

**Assumption H.4** (Bounded global gradient variance). There exists a constant $\sigma_g^2 > 0$ such that the global gradient variance is bounded as follows: $\frac{1}{U}\sum_{u=1}^{U}\|\nabla \mathcal{L}_u(w) - \nabla \mathcal{L}(w)\|^2 \leq \sigma_g^2$.

## H.2    PRELIMINARIES

**Lemma H.5** (Jensen's Inequality). *For any $a_u \in \mathbb{R}^d, u \in \{1, 2, \ldots, U\}$:*

$$\|\frac{1}{U}\sum_{u=1}^{U}\mathbf{a}_u\|^2 \leq \frac{1}{U}\sum_{u=1}^{U}\|\mathbf{a}_u\|^2 \tag{7}$$

$$\|\sum_{u=1}^{U}\mathbf{a}_u\|^2 \leq U\sum_{u=1}^{U}\|\mathbf{a}_u\|^2 \tag{8}$$

**Lemma H.6** ((Khaled et al., 2020)). *If $\mathcal{L}$ is smooth and convex, then*

$$\|\nabla \mathcal{L}(w) - \nabla \mathcal{L}(w')\|^2 \leq 2L(\mathcal{L}(w) - \mathcal{L}(w') - \langle \nabla \mathcal{L}(w'), w - w'\rangle) \tag{9}$$

**Lemma H.7** (Co-coercivity of convex $L$-smooth function). *If $F$ is $L$-smooth and convex then*

$$\langle \nabla \mathcal{L}(w) - \nabla \mathcal{L}(w'), w - w'\rangle \geq \frac{1}{L}\|\nabla \mathcal{L}(w) - \nabla \mathcal{L}(w')\|^2. \tag{10}$$

*A direct consequence of this lemma is:*

$$\langle \nabla \mathcal{L}(w), w - w^*\rangle \geq \frac{1}{L}\|\nabla \mathcal{L}(w)\|^2, \tag{11}$$

*where $w^*$ is a minimizer of $\mathcal{L}(w)$*

**Lemma H.8** (Co-coercivity of $\mu$-strongly convex function). *If $F$ is $L$-smooth and convex then*

$$\langle \nabla \mathcal{L}(w) - \nabla \mathcal{L}(w'), w - w'\rangle \geq \frac{1}{L}\|\nabla \mathcal{L}(w) - \nabla \mathcal{L}(w')\|^2. \tag{12}$$

*A direct consequence of this lemma is:*

$$\langle \nabla \mathcal{L}(w), w - w^*\rangle \geq \frac{1}{L}\|\nabla \mathcal{L}(w)\|^2, \tag{13}$$

*where $w^*$ is a minimizer of $F(w)$*

### H.3 Reformulation for Vanilla pFL and FedLAG

In the appendix, we aim to achieve two main objectives. Firstly, we intend to illustrate the distinction between FedLAG and other Federated Learning (FL) algorithms that do not incorporate FedLAG. Secondly, we aim to demonstrate the convergence under novel FedLAG settings. To accomplish these goals, we introduce additional notations in the appendix. It is essential to note that these extensively defined notations are exclusive to the appendix and do not compromise the generality of the main paper. Towards the conclusion of the appendix, our objective is to consolidate the formulations and revert the convergence formulation back to its original form.

We define $w_u^{(r)}, w_g^{(r)}$ as local and global model parameters, respectively. In the appendix, to distinguish between two concepts Vanilla pFL and FedLAG (with our proposed layer-wise personalization technique), we also define the following auxiliary variables that will used in the proof:

- **Layer-wise Personalized Model:** To represent the layer disentanglement architecture, where the model are layer-wise personalized, we define the user's model $w_{\text{LAG},u}^{(r)} = \{\theta_{s,u}^{(r)}, \theta_{p,u}^{(r)}\}$, where $\{\theta_{s,u}^{(r)}, \theta_{p,u}^{(r)}\}$ represents the model layers that is assigned for global aggregation, and the one that is assigned for personalized purpose, respectively.

- **Vanilla FL Model:** We define the user's model $w_{\text{VFL},u}^{(r)} = \{\theta_{s,u}^{(r)}, \theta_{q,u}^{(r)}\}$, where $\{\theta_{s,u}^{(r)}, \theta_{q,u}^{(r)}\}$ represents the model layers that is assigned for global aggregation, and the one that is assigned for personalized purpose. However, in Vanilla settings, we do the shared global aggregation on the personalized model. Thus, the model update at global aggregation is similar to the shared layers.

As the model layer $\theta_{s,u}^{(r)}$ is assigned for the global aggregation. Without the loss of generality, the two model layers have same attributes and update rules. Consequently, we use the same notations $\theta_{s,u}^{(r)}$ for the two model layers according to **Layer-wise Personalized Model** and **Vanilla FL Model**.

- **Accumulated Local Gradient:** We have the local gradient update on the whole user's network:

$$h_u^{(r)} = \sum_{e=0}^{E-1} \eta \nabla \mathcal{L}_u(w_u^{(r,e)}) = w_u^{(r,E)} - w_u^{(r)}. \tag{14}$$

By decomposing into shared aggregation layer set $\theta_{s,u}^{(r)}$, and personalized layer set $\theta_{p,u}^{(r)}$, we have the following:

$$h_{s,u}^{(r)} = \sum_{e=0}^{E-1} \eta \nabla \mathcal{L}_u(\theta_{s,u}^{(r,e)}) = \theta_{s,u}^{(r,E)} - \theta_{s,u}^{(r)}, \tag{15}$$

$$h_{p,u}^{(r)} = \sum_{e=0}^{E-1} \eta \nabla \mathcal{L}_u(\theta_{p,u}^{(r,e)}) = \theta_{p,u}^{(r,E)} - \theta_{p,u}^{(r)}. \tag{16}$$

As the local update on both vanilla and FedLAG remain the same, thus, we have the gradient trajectories on both setting similar. For instance,

$$h_{p,u}^{(r)} = h_{q,u}^{(r)}. \tag{17}$$

- **Global Gradient Aggregation:** For FedLAG, we have the two different update rules for two different type of layers:

$$\theta_{s,u}^{(r+1,0)} = \frac{1}{U} \sum_{u=1}^{U} \theta_{s,u}^{(r)} = \theta_{s,u}^{(r,0)} - \frac{1}{U} \sum_{u=1}^{U} h_{s,u}^{(r)},$$

$$\theta_{p,u}^{(r+1,0)} = \frac{1}{U} \sum_{u=1}^{U} \theta_{p,u}^{(r)} = \theta_{p,u}^{(r,0)} - h_{p,u}^{(r)}. \tag{18}$$

On Vanilla pFL, we have the same update rule for two different type of layers:

$$\theta_{su}^{(r+1,0)} = \frac{1}{U}\sum_{u=1}^{U}\theta_{s,u}^{(r)} = \theta_{s,0}^{(r,0)} - \frac{1}{U}\sum_{u=1}^{U}h_{s,u}^{(r)}, \tag{19}$$

$$\theta_{q,u}^{(r+1,0)} = \frac{1}{U}\sum_{u=1}^{U}\theta_{p,u}^{(r)} = \theta_{p,u}^{(r,0)} - \frac{1}{U}\sum_{u=1}^{U}h_{p,u}^{(r)}. \tag{20}$$

- **Loss Function:** Due to the definition of two models $\theta_{s,u}, \theta_{p,u}$, we can define the $\mathcal{L}_u(\theta_{s,g}^{(r)}), \mathcal{L}_u(\theta_{p,g}^{(r)})$ are the two empirical loss that aim to design a global optimal models $w^* = \{\theta_{s,u}^*, \theta_{p,u}^*\}$, respectively. Therefore, we have $\mathcal{L}_u(w_{\text{LAG},u}^{(r)}) = \mathcal{L}_u(\theta_{s,g}^{(r)}) + \mathcal{L}_u(\theta_{p,g}^{(r)})$.

## H.4 PROOF ON LEMMA 5.1

By leveraging Taylor approximation, we have the following loss estimation for joint model parameters at each communication round. For the layer-wise personalized aggregation, we have:

$$\mathcal{L}_u(\theta_{s,u}^{(r+1)}, \theta_{p,u}^{(r+1)}) \overset{(a)}{=} \mathcal{L}_u(\theta_{s,u}^{(r+1)}, \theta_{p,u}^{(r)}) + (\theta_{p,u}^{(r+1)} - \theta_{p,u}^{(r)})\nabla_{\theta_{s,u}^{(r)}}\mathcal{L}_u(\theta_{s,u}^{(r)}, \theta_{p,u}^{(r)}) + \mathcal{O}(\eta)$$

$$\overset{(b)}{=} \mathcal{L}_u(\theta_{s,u}^{(r)}, \theta_{p,u}^{(r)}) + (\theta_{s,u}^{(r+1)} - \theta_{s,u}^{(r)})\nabla_{\theta_{s,u}^{(r)}}\mathcal{L}_u(\theta_{s,u}^{(r)}, \theta_{p,u}^{(r)})$$

$$+ (\theta_{p,u}^{(r+1)} - \theta_{p,u}^{(r)})\nabla_{\theta_{p,u}^{(r)}}\mathcal{L}_u(\theta_{s,u}^{(r)}, \theta_{p,u}^{(r)}) + \mathcal{O}(\eta)$$

$$= \mathcal{L}_u(\theta_{s,u}^{(r)}, \theta_{p,u}^{(r)}) + (\theta_{s,u}^{(r+1)} - \theta_{s,u}^{(r)})\sum_{e=0}^{E-1}\eta\nabla_{\theta_{s,u}^{(r,e)}}\mathcal{L}_u(\theta_{s,u}^{(r,e)}, \theta_{p,u}^{(r,e)})$$

$$+ (\theta_{p,u}^{(r+1)} - \theta_{p,u}^{(r)})\sum_{e=0}^{E-1}\eta\nabla_{\theta_{p,u}^{(r,e)}}\mathcal{L}_u(\theta_{s,u}^{(r,e)}, \theta_{p,u}^{(r,e)})$$

$$= \mathcal{L}_u(\theta_{s,u}^{(r)}, \theta_{p,u}^{(r)}) + (\theta_{s,u}^{(r+1)} - \theta_{s,u}^{(r)})h_{s,u}^{(r)} + (\theta_{p,u}^{(r+1)} - \theta_{p,u}^{(r)})h_{p,u}^{(r)} + \mathcal{O}(\eta), \tag{21}$$

where (a) and (b) hold due to the approximations of $\mathcal{L}_u(\theta_{s,u}^{(r+1)}, \theta_{p,u}^{(r+1)})$ according to $\theta_{s,u}^{(r+1)}$ and $\theta_{p,u}^{(r+1)}$, respectively. For normal update, we have:

$$\mathcal{L}_u(\theta_{s,u}^{(r+1)}, \theta_{q,u}^{(r+1)}) \overset{(a)}{=} \mathcal{L}_u(\theta_{s,u}^{(r)}, \theta_{q,u}^{(r+1)}) + (\theta_{q,u}^{(r+1)} - \theta_{q,u}^{(r)})\nabla_{\theta_{s,u}^{(r)}}\mathcal{L}_u(\theta_{s,u}^{(r)}, \theta_{q,u}^{(r)}) + \mathcal{O}(\eta)$$

$$\overset{(b)}{=} \mathcal{L}_u(\theta_{s,u}^{(r)}, \theta_{q,u}^{(r)}) + (\theta_{s,u}^{(r+1)} - \theta_{s,u}^{(r)})\nabla_{\theta_{s,u}^{(r)}}\mathcal{L}_u(\theta_{s,u}^{(r)}, \theta_{q,u}^{(r)})$$

$$+ (\theta_{q,u}^{(r+1)} - \theta_{q,u}^{(r)})\nabla_{\theta_{q,u}^{(r)}}\mathcal{L}_u(\theta_{s,u}^{(r)}, \theta_{q,u}^{(r)}) + \mathcal{O}(\eta)$$

$$= \mathcal{L}_u(\theta_{s,u}^{(r)}, \theta_{q,u}^{(r)}) + (\theta_{s,u}^{(r+1)} - \theta_{s,u}^{(r)})\sum_{e=0}^{E-1}\eta\nabla_{\theta_{s,u}^{(r,e)}}\mathcal{L}_u(\theta_{s,u}^{(r,e)}, \theta_{q,u}^{(r,e)})$$

$$+ (\theta_{q,u}^{(r+1)} - \theta_{q,u}^{(r)})\sum_{e=0}^{E-1}\eta\nabla_{\theta_{q,u}^{(r,e)}}\mathcal{L}_u(\theta_{s,u}^{(r,e)}, \theta_{q,u}^{(r,e)})$$

$$= \mathcal{L}_u(\theta_{s,u}^{(r)}, \theta_{q,u}^{(r)}) + (\theta_{s,u}^{(r+1)} - \theta_{s,u}^{(r)})h_{s,u}^{(r)} + (\theta_{q,u}^{(r+1)} - \theta_{q,u}^{(r)})h_{q,u}^{(r)} + \mathcal{O}(\eta)$$

$$\overset{(c)}{=} \mathcal{L}_u(\theta_{s,u}^{(r)}, \theta_{q,u}^{(r)}) + (\theta_{s,u}^{(r+1)} - \theta_{s,u}^{(r)})h_{s,u}^{(r)} + (\theta_{q,u}^{(r+1)} - \theta_{q,u}^{(r)})h_{p,u}^{(r)} + \mathcal{O}(\eta), \tag{22}$$

where $(a)$ and $(b)$ hold due to the approximations of $\mathcal{L}_u(\theta_{s,u}^{(r+1)}, \theta_{q,u}^{(r+1)})$ according to $\theta_{s,u}^{(r+1)}$ and $\theta_{q,u}^{(r+1)}$, respectively. $(c)$ holds according to the Equation 17. Specifically, in the one-step layer-wise personalized aggregation, we want to estimate the divergence that loss function from LAG move from the vanilla pFL. The difference between the two loss function after the update is measured by subtracting $\mathcal{L}_u(\theta_{s,u}^{(r+1)}, \theta_{p,u}^{(r+1)})$ in Equation equation 21 from $\mathcal{L}_u(\theta_{s,u}^{(r+1)}, \theta_{q,u}^{(r+1)})$ in Equation

equation 22 as follows:

$$
\begin{aligned}
&\mathcal{L}_u(\theta_{s,u}^{(r+1)}, \theta_{p,u}^{(r+1)}) - \mathcal{L}_u(\theta_{s,u}^{(r+1)}, \theta_{q,u}^{(r+1)}) \\
&= \left( \mathcal{L}_u(\theta_{s,u}^{(r)}, \theta_{p,u}^{(r)}) + (\theta_{s,u}^{(r+1)} - \theta_{s,u}^{(r)}) h_{s,u}^{(r)} + (\theta_{p,u}^{(r+1)} - \theta_{p,u}^{(r)}) h_{p,u}^{(r)} \right) \\
&\quad - \left( \mathcal{L}_u(\theta_{s,u}^{(r)}, \theta_{q,u}^{(r)}) + (\theta_{s,u}^{(r+1)} - \theta_{s,u}^{(r)}) h_{s,u}^{(r)} + (\theta_{q,u}^{(r+1)} - \theta_{q,u}^{(r)}) h_{p,u}^{(r)} \right) + \mathcal{O}(\eta) \\
&= (\theta_{p,u}^{(r+1)} - \theta_{q,u}^{(r+1)}) h_{p,u}^{(r)} + (\theta_{p,u}^{(r)} - \theta_{q,u}^{(r)}) h_{p,u}^{(r)} + \mathcal{O}(\eta), \\
&\stackrel{(a)}{=} (\theta_{p,u}^{(r+1)} - \theta_{q,u}^{(r+1)}) h_{p,u}^{(r)} + \mathcal{O}(\eta), \\
&= -\eta (h_{p,u}^{(r)} - \frac{1}{U} \sum_{v=1}^{U} h_{p,v}^{(r)})^\top h_{p,u}^{(r)} + \mathcal{O}(\eta), \\
&= -\eta \frac{1}{U} \sum_{v=1}^{U} (h_{p,u}^{(r)} - h_{p,v}^{(r)})^\top h_{p,u}^{(r)} + \mathcal{O}(\eta), \\
&= -\eta \frac{1}{U} \sum_{v=1}^{U} (\|h_{p,u}^{(r)}\|^2 - h_{p,v}^{(r)\top} h_{p,u}^{(r)}) + \mathcal{O}(\eta),
\end{aligned}
\tag{23}
$$

where $\mathcal{O}(\eta)$ represents the remainder which depends on the variable $\eta$. $(a)$ holds due to the model are considered to be trained at the same starting position at each round $r$, thus $\theta_{p,u}^{(r)} = \theta_{q,u}^{(r)}$. Without loss of generality, we assume that $\|h_{p,u}^{(r)}\| \neq 0$, and $h_{p,u}^{(r)} = \{h_{l,u}^{(r)} | \forall l \in \mathbb{L}_p\}$ then:

$$
\begin{aligned}
\|h_{p,u}^{(r)}\|^2 - h_{p,v}^{(r)\top} h_{p,u}^{(r)} &= \sum_{l \in \mathbb{L}_p} (\|h_{l,u}^{(r)}\|^2 - h_{l,v}^{(r)\top} h_{l,u}^{(r)}), \\
&= \sum_{l \in \mathbb{L}_p} \|h_{l,u}^{(r)}\| (\|h_{l,u}^{(r)}\| - \cos \Phi_{u,v}^{(l)} \|h_{l,v}^{(r)}\|) \stackrel{(a)}{>} 0,
\end{aligned}
\tag{24}
$$

where $\mathbb{L}_p$ represents the layers that have gradient conflict and turned become personalized layers. Due to the gradient conflict definition which has proposed in Definition 2.1, we have: $\cos \Phi_{u,v}^{(l)} < 0$ as $\Phi_{u,v}^{(l)} > \pi/2$, which makes the inequality (a) hold. Hence, the above difference is negative, if $\eta$ is sufficiently small. As such, the difference between the vanilla FL and LAG loss functions is also negative, if $\eta$ is sufficiently small.

## H.5 Proof on Lemma 6

From Lemma 5.1, we have

$$
\begin{aligned}
\mathcal{L}(\theta_{s,g}^{(r+1)}, \theta_{p,g}^{(r+1)}) - \mathcal{L}(\theta_{s,g}^{(r+1)}, \theta_{q,g}^{(r+1)}) &= \frac{1}{U}\sum_{u=1}^{U}\mathcal{L}_u(\theta_{s,u}^{(r+1)}, \theta_{p,u}^{(r+1)}) - \frac{1}{U}\sum_{u=1}^{U}\mathcal{L}_u(\theta_{s,u}^{(r+1)}, \theta_{q,u}^{(r+1)}) \\
&= -\eta\frac{1}{U^2}\sum_{u=1}^{U}\sum_{v=1}^{U}\left(\|h_{p,u}^{(r)}\|^2 - h_{p,v}^{(r)\top}h_{p,u}^{(r)}\right) \\
&\overset{(a)}{=} -\eta\frac{1}{U^2}\sum_{u=1}^{U}\sum_{v=1}^{U}\sum_{l\in\mathbb{L}}\left(\|h_{l,u}^{(r)}\|^2 - h_{l,v}^{(r)}\cdot h_{l,u}^{(r)}\right) \\
&\overset{(b)}{=} -\eta\frac{1}{U^2}\sum_{u=1}^{U}\sum_{v=1}^{U}\sum_{l\in\mathbb{L}_p}\left(\|h_{l,u}^{(r)}\|^2 - \frac{(h_{l,u}^{(r)})\cdot h_{l,v}^{(r)}}{\|h_{l,u}^{(r)}\|\|h_{l,v}^{(r)}\|}\|h_{l,u}^{(r)}\|\|h_{l,v}^{(r)}\|\right) \\
&= -\eta\frac{1}{U^2}\sum_{u=1}^{U}\sum_{v=1}^{U}\sum_{l\in\mathbb{L}_p}\|h_{l,u}^{(r)}\|\left(\|h_{l,u}^{(r)}\| - \frac{h_{l,u}^{(r)}\cdot h_{l,v}^{(r)}}{\|h_{l,u}^{(r)}\|\|h_{l,v}^{(r)}\|}\|h_{l,v}^{(r)}\|\right) \\
&\overset{(c)}{=} -\eta\frac{1}{U^2}\sum_{u=1}^{U}\sum_{v=1}^{U}\sum_{l\in\mathbb{L}_p}\|h_{l,u}^{(r)}\|\left(\|h_{l,u}^{(r)}\| - \cos\Phi_{u,v}^{(l)}\|h_{l,v}^{(r)}\|\right) < 0,
\end{aligned}
\tag{25}
$$

where $(a)$ holds according to the Lemma H.9 and $(b)$ holds as the difference only be made on the personalized layers $\theta_{p,u}^{(r)}$. $(c)$ holds according to Definition 2.1

**Lemma H.9.** *Given two vector $h_{p,v}^{(r)}, h_{p,u}^{(r)} \in \mathbb{R}^P$, we have following relationship:*

$$
h_{p,v}^{(r)\top}h_{p,u}^{(r)} = h_{p,v}^{(r)}\cdot h_{p,u}^{(r)} = \sum_{l=1}^{\mathbb{L}}h_{l,u}^{(r)}\cdot h_{l,v}^{(r)},
\tag{26}
$$

*where $P = \sum_{l=1}^{L}P_l$ is the number of parameters of the FL model, $P_l$ is the number of parameters on each layer $l$.*

## H.6 Upper boundary on Layer-wise Loss Improvement Approximation

Consider the inequality in Eq. equation 25. To find an upper boundary on the loss improvement, we consider the maximal gradient norm $\widetilde{h}_{p,u}^{(r)} = \arg\max_{h_{l,u}^{(r)}}\|h_{l,u}^{(r)}\|$. Thus, we have:

$$
\begin{aligned}
\mathcal{L}(\theta_{s,g}^{(r+1)}, \theta_{p,g}^{(r+1)}) - \mathcal{L}(\theta_{s,g}^{(r+1)}, \theta_{q,g}^{(r+1)}) &= -\eta\frac{1}{U^2}\sum_{u=1}^{U}\sum_{v=1}^{U}\sum_{l\in\mathbb{L}_p}\|h_{l,u}^{(r)}\|\left(\|h_{l,u}^{(r)}\| - \cos\Phi_{u,v}^{(l)}\|h_{l,v}^{(r)}\|\right) \\
&= -\eta\mathbb{E}_{u,v}\left[\sum_{l\in\mathbb{L}_p}\|h_{l,u}^{(r)}\|\left(\|h_{l,u}^{(r)}\| - \cos\Phi_{u,v}^{(l)}\|h_{l,v}^{(r)}\|\right)\right] \\
&\leq -\eta\mathbb{E}_{u,v}\left[\sum_{l\in\mathbb{L}_p}\|\widetilde{h}_{p,u}^{(r)}\|\left(\|\widetilde{h}_{p,u}^{(r)}\| - \cos\Phi_{u,v}^{(l)}\|\widetilde{h}_{p,v}^{(r)}\|\right)\right] \\
&= -\eta\mathbb{E}_{u,v}\left[L_p\|\widetilde{h}_{p,u}^{(r)}\|\left(\|\widetilde{h}_{p,u}^{(r)}\| - \cos\Phi_{u,v}^{(l)}\|\widetilde{h}_{p,v}^{(r)}\|\right)\right] \\
&= -\eta\mathbb{E}_{u,v}\left[\frac{L_p}{L^2}\|\widetilde{h}_{u}^{(r)}\|\left(\|\widetilde{h}_{u}^{(r)}\| - \cos\Phi_{u,v}^{(l)}\|\widetilde{h}_{v}^{(r)}\|\right)\right] \\
&\leq -\eta\mathbb{E}_{u,v}\left[\|\widetilde{h}_{u}^{(r)}\|\left(\|\widetilde{h}_{u}^{(r)}\| - \cos\Phi_{u,v}^{(l)}\|\widetilde{h}_{v}^{(r)}\|\right)\right].
\end{aligned}
\tag{27}
$$

Inequality in Eq. equation 27 proves that the improvement of layer-wise loss is upper-bounded by the model gradient norm. Furthermore, the improvement on AI model depends on the percentage of layers that required to be personalized rather than that of the number of layers. As a consequence, the formula is agnostic to the over-parameterized of the AI model.

### H.7 BOUNDING USER AGGREGATE GRADIENTS

**Lemma H.10** ((Jhunjhunwala et al., 2023), Bounding user aggregate gradients on Vanilla FL).

$$\frac{1}{U} \sum_{u=1}^{U} \sum_{e=0}^{E-1} \|\nabla\mathcal{L}_u(w_{s,u}^{(r,e)})\|^2 \le \frac{3L^2}{U} \sum_{u=1}^{U} \sum_{e=0}^{E-1} \|w_{s,g}^{(r,e)} - w_{VFL,g}^{(r)}\|^2 \tag{28}$$

$$+ 6EL(\nabla\mathcal{L}(w_{VFL,g}^{(r)}) - \nabla\mathcal{L}(w_g^*)) + 3E\sigma_*^2. \tag{29}$$

*The lemma shows that the local gradient variance $\mathcal{L}_u(w_{s,u}^{(r,e)})$ on user $u$ after $E$ local epochs is always bounded by a certain threshold.*

**Lemma H.11** (Bounding user aggregate gradients on Layer-wise Personalized FL).

$$\frac{1}{U} \sum_{u=1}^{U} \sum_{e=0}^{E-1} \|\nabla\mathcal{L}_u(w_{LAG,u}^{(r,e)})\|^2 \le \frac{3L^2}{U} \sum_{u=1}^{U} \sum_{e=0}^{E-1} \|w_{LAG,g}^{(r,e)} - w_{LAG,g}^{(r)}\|^2 + 6EL\Big(\mathcal{L}(w^{(r)} - \mathcal{L}(w^*))\Big)$$

$$- \frac{6EL\eta}{U^2} \sum_{u=1}^{U} \sum_{v=1}^{U} \sum_{l\in\mathbb{L}_p} \|h_{l,u}^{(r)}\|\Big(\|h_{l,u}^{(r)}\| - \cos\Phi_{u,v}^{(l)}\|h_{l,v}^{(r)}\|\Big) + 3E\sigma_*^2. \tag{30}$$

*The lemma shows that the local gradient variance $\mathcal{L}_u(w_u^{(r,e)})$ on user $u$ after $E$ local epochs is always bounded by a certain threshold.*

**Proof:**

$$\frac{1}{U} \sum_{u=1}^{U} \sum_{e=0}^{E-1} \|\nabla\mathcal{L}_u(w_{LAG,u}^{(r,e)})\|^2$$

$$= \frac{1}{U} \sum_{u=1}^{U} \sum_{e=0}^{E-1} \|\nabla\mathcal{L}_u(w_{LAG,u}^{(r,e)}) - \nabla\mathcal{L}_u(w_{LAG,u}^{(r)}) + \nabla\mathcal{L}_u(w_{LAG,u}^{(r)}) - \nabla\mathcal{L}_u(w^*) + \nabla\mathcal{L}_u(w^*)\|^2$$

$$\le \underbrace{\frac{3}{U} \sum_{u=1}^{U} \sum_{e=0}^{E-1} \|\nabla\mathcal{L}_u(w_{LAG,u}^{(r,e)}) - \nabla\mathcal{L}_u(w_{LAG,u}^{(r)})\|^2}_{Q_1} + \underbrace{\frac{3}{U} \sum_{u=1}^{U} \sum_{e=0}^{E-1} \|\nabla\mathcal{L}_u(w_{LAG,u}^{(r)}) - \nabla\mathcal{L}_u(w^*)\|^2}_{Q_2}$$

$$\tag{31}$$

$$+ \underbrace{\frac{3}{U} \sum_{u=1}^{U} \sum_{e=0}^{E-1} \|\nabla\mathcal{L}_u(w^*)\|^2}_{Q_3}. \tag{32}$$

We derive $Q_1, Q_2, Q_3$ as follows:

$$Q_1 = \frac{3}{U} \sum_{u=1}^{U} \sum_{e=0}^{E-1} \|\nabla\mathcal{L}_u(w_{LAG,u}^{(r,e)}) - \nabla\mathcal{L}_u(w_{LAG,u}^{(r)})\|^2 \le \frac{3L^2}{U} \sum_{u=1}^{U} \sum_{e=0}^{E-1} \|w_{LAG,g}^{(r,e)} - w_{LAG,g}^{(r)}\|^2 \tag{33}$$

$$Q_2 = \frac{3}{U} \sum_{u=1}^{U} \sum_{e=0}^{E-1} \|\nabla \mathcal{L}_u(w_{\text{LAG},u}^{(r)}) - \nabla \mathcal{L}_u(w^*)\|^2$$

$$\leq \frac{3}{U} \sum_{u=1}^{U} \sum_{e=0}^{E-1} \Big[ 2L \Big( \nabla \mathcal{L}_u(w_{\text{LAG},u}^{(r)}) - \nabla \mathcal{L}_u(w^*) \Big) - \langle \nabla \mathcal{L}_u(w^*), w_{\text{LAG},u}^{(r)} - w^* \rangle \Big]$$

$$\overset{(a)}{=} \frac{6L}{U} \sum_{u=1}^{U} \sum_{e=0}^{E-1} \Big( \nabla \mathcal{L}_u(w_{\text{LAG},u}^{(r)}) - \nabla \mathcal{L}_u(w^*) \Big)$$

$$= 6EL \Big( \frac{1}{U} \sum_{u=1}^{U} \nabla \mathcal{L}_u(w_{\text{LAG},u}^{(r)}) - \frac{1}{U} \sum_{u=1}^{U} \nabla \mathcal{L}_u(w^*) \Big) = 6EL \Big( \nabla \mathcal{L}(w_{\text{LAG},g}^{(r)}) - \nabla \mathcal{L}(w^*) \Big) \quad (34)$$

$$Q_3 = \frac{3}{U} \sum_{u=1}^{U} \sum_{e=0}^{E-1} \|\nabla \mathcal{L}_u(w^*)\|^2 \leq 3E\sigma_*^2 \quad (35)$$

Therefore, we have:

$$\frac{1}{U} \sum_{u=1}^{U} \sum_{e=0}^{E-1} \|\nabla \mathcal{L}_u(w_u^{(r,e)})\|^2$$

$$\leq \frac{3L^2}{U} \sum_{u=1}^{U} \sum_{e=0}^{E-1} \|w_{\text{LAG},g}^{(r,e)} - w_{\text{LAG},g}^{(r)}\|^2 + 6EL \Big( \mathcal{L}(w_{\text{LAG},g}^{(r)}) - \mathcal{L}(w^*) \Big) + 3E\sigma_*^2. \quad (36)$$

Base on H.4, we have:

$$\frac{1}{U} \sum_{u=1}^{U} \sum_{e=0}^{E-1} \|\nabla \mathcal{L}_u(w_{\text{LAG},u}^{(r,e)})\|^2$$

$$\leq \frac{3L^2}{U} \sum_{u=1}^{U} \sum_{e=0}^{E-1} \|w_{\text{LAG},g}^{(r,e)} - w_{\text{LAG},g}^{(r)}\|^2 + 6EL \Big( \mathcal{L}(w^{(r)} - \mathcal{L}(w^*) \Big)$$

$$- \frac{6EL\eta}{U^2} \sum_{u=1}^{U} \sum_{v=1}^{U} \sum_{l \in \mathbb{L}_p} \|h_{l,u}^{(r)}\| \Big( \|h_{l,u}^{(r)}\| - \cos \Phi_{u,v}^{(l)} \|h_{l,v}^{(r)}\| \Big) + 3E\sigma_*^2. \quad (37)$$

## H.8 BOUNDING USER DRIFT

**Lemma H.12** ((Jhunjhunwala et al., 2023), Bounding user drift on Vanilla FL).

$$\frac{1}{U}\sum_{u=1}^{U}\sum_{e=0}^{E-1}\|w_u^{(r)} - w_u^{(r,e)}\|^2 \leq 12\eta^2 E^2(E-1)L(F(w^{(r)}) - F(w^*)) + 6\eta^2 E^2(E-1)\sigma_*^2. \quad (38)$$

**Lemma H.13** (Bounding user drift on Layer-wise Personalized FL).

$$\frac{1}{U}\sum_{u=1}^{U}\sum_{e=0}^{E-1}\|w_{LAG,u}^{(r)} - w_{LAG,u}^{(r,e)}\|^2 \leq 12\eta^2 E^2(E-1)L(F(w^{(r)}) - F(w^*)) + 6\eta^2 E^2(E-1)\sigma_*^2.$$
$$\quad (39)$$

**Proof:**

$$\frac{1}{U}\sum_{u=1}^{U}\sum_{e=0}^{E-1}\|w_{\text{LAG},u}^{(r)} - w_{\text{LAG},u}^{(r,e)}\|^2$$

$$= \eta^2\frac{1}{U}\sum_{u=1}^{U}\sum_{e=0}^{E-1}\|\sum_{i=0}^{e}\nabla\mathcal{L}_u(w_{\text{LAG},u}^{r,i})\|^2$$

$$\leq \eta^2\frac{1}{U}\sum_{u=1}^{U}\sum_{e=0}^{E-1}e\sum_{i=0}^{e}\|\nabla\mathcal{L}_u(w_{\text{LAG},u}^{r,i})\|^2$$

$$\leq \eta^2 E(E-1)\frac{1}{U}\sum_{u=1}^{U}\sum_{e=0}^{E-1}\|\nabla\mathcal{L}_u(w_{\text{LAG},u}^{r,e})\|^2$$

$$\overset{(a)}{\leq} \eta^2 E(E-1)\Big[\frac{3L^2}{U}\sum_{u=1}^{U}\sum_{e=0}^{E-1}\|w_{\text{LAG},g}^{(r,e)} - w_{\text{LAG},g}^{(r)}\|^2 + 6EL\Big(\mathcal{L}(w^{(r)} - \mathcal{L}(w^*))\Big)$$

$$- \frac{6EL\eta}{U^2}\sum_{u=1}^{U}\sum_{v=1}^{U}\sum_{l\in\mathbb{L}_p}\|h_{l,u}^{(r)}\|\Big(\|h_{l,u}^{(r)}\| - \cos\Phi_{u,v}^{(l)}\|h_{l,v}^{(r)}\|\Big) + 3E\sigma_*^2\Big]$$

$$\overset{(b)}{\leq} \frac{1}{2U}\sum_{u=1}^{U}\sum_{e=0}^{E-1}\|w_{\text{LAG},g}^{(r,e)} - w_{\text{LAG},g}^{(r)}\|^2 + 6E^2(E-1)\eta^2 L\Big(\mathcal{L}(w^{(r)} - \mathcal{L}(w^*))\Big)$$

$$- 6E^2(E-1)\eta^3\frac{L}{U^2}\sum_{u=1}^{U}\sum_{v=1}^{U}\sum_{l\in\mathbb{L}_p}\|h_{l,u}^{(r)}\|\Big(\|h_{l,u}^{(r)}\| - \cos\Phi_{u,v}^{(l)}\|h_{l,v}^{(r)}\|\Big) + 3\eta^2 E^2(E-1)\sigma_*^2, \quad (40)$$

where (a) holds due to the Lemma H.11 and we have (b) due to the assumption with constraints $\eta^2 \leq \frac{1}{6EL}$. Therefore, we have:

$$\frac{1}{U}\sum_{u=1}^{U}\sum_{e=0}^{E-1}\|w_{\text{LAG},u}^{(r)} - w_{\text{LAG},u}^{(r,e)}\|^2 \leq 12E^2(E-1)\eta^2 L\Big(\mathcal{L}(w^{(r)} - \mathcal{L}(w^*))\Big)$$

$$- 12E^2(E-1)\eta^3\frac{L}{U^2}\sum_{u=1}^{U}\sum_{v=1}^{U}\sum_{l\in\mathbb{L}_p}\|h_{l,u}^{(r)}\|\Big(\|h_{l,u}^{(r)}\| - \cos\Phi_{u,v}^{(l)}\|h_{l,v}^{(r)}\|\Big) + 6\eta^2 E^2(E-1)\sigma_*^2$$
$$\quad (41)$$

## H.9 Proof on Theorem 5.3

We define $\bar{h}^{(r)} = \{\bar{h}_s^{(r)}, \bar{h}_p^{(r)}\}$, where $\bar{h}_s^{(r)}, \bar{h}_p^{(r)}$ represents the gradient update rules for shared layers and personalized layers, respectively. Recall that the update of the global model can be written as $w_{\text{LAG},g}^{(r+1)} = w_{\text{LAG},g}^{(r)} - \eta \frac{1}{U} \sum_{u=1}^{U} \bar{h}_u^{(r)}$. When disentangle into two distinguished models, we have the update for shared layers as $\theta_{s,g}^{(r+1)} = \theta_{s,g}^{(r)} - \eta \frac{1}{U} \sum_{u=1}^{U} \bar{h}_{s,u}^{(r)}$ and personalized layers as $\theta_{p,g}^{(r+1)} = \theta_{p,u}^{(r)} - \eta \bar{h}_{p,u}^{(r)}$. Therefore, we have:

$$
\mathbb{E}\left[\|w_{\text{LAG},g}^{(r+1)} - w^*\|^2\right] = \mathbb{E}_{u \in U}\left[\|\theta_{s,g}^{(r+1)} - \theta_{s,g}^*\|^2\right] + \mathbb{E}_{u \in U}\left[\|\theta_{p,u}^{(r+1)} - \theta_{p,g}^*\|^2\right]
$$

$$
= \|\theta_{s,g}^{(r)} - \frac{\eta}{U} \sum_{u=1}^{U} h_{s,u}^{(r)} - \theta_{s,g}^*\|^2 + \mathbb{E}_{u \in U}\left[\|\theta_{p,u}^{(r)} - \eta h_{p,u}^{(r)} - \theta_{p,g}^*\|^2\right]
$$

$$
= \|\theta_{s,g}^{(r)} - \theta_{s,g}^*\|^2 + \mathbb{E}_{u \in U}\left[\|\theta_{p,u}^{(r)} - \theta_{p,g}^*\|^2\right] - 2\eta \langle \theta_{s,g}^{(r)} - \theta_{s,g}^*, \frac{1}{U} \sum_{u=1}^{U} h_{s,u}^{(r)} \rangle
$$

$$
- 2\eta \mathbb{E}_{u \in U}\left[\langle \theta_{p,u}^{(r)} - \theta_{p,g}^*, h_{p,u}^{(r)} \rangle\right] + \|\frac{\eta}{U} \sum_{u=1}^{U} h_{s,u}^{(r)}\|^2 + \mathbb{E}_{u \in U}\left[\|\eta h_{p,u}^{(r)}\|^2\right]
$$

$$
\leq \|w_{\text{LAG},g}^{(r)} - w^*\|^2 - 2\eta \underbrace{\langle \theta_{s,g}^{(r)} - \theta_{s,g}^*, \frac{1}{U} \sum_{u=1}^{U} h_{s,u}^{(r)} \rangle}_{Q_1} + \eta^2 \underbrace{\frac{1}{U} \sum_{u=1}^{U} \|h_{s,u}^{(r)}\|^2}_{Q_3}
$$

$$
- 2\eta \underbrace{\frac{1}{U} \sum_{u=1}^{U} \langle \theta_{p,u}^{(r)} - \theta_{p,g}^*, h_{p,u}^{(r)} \rangle}_{Q_2} + \eta^2 \underbrace{\frac{1}{U} \sum_{u=1}^{U} \|h_{p,u}^{(r)}\|^2}_{Q_4} \tag{42}
$$

**Bounding $Q_1$:** We have:

$$
Q_1 = \langle \theta_{s,g}^{(r)} - \theta_{s,g}^*, \frac{1}{U} \sum_{u=1}^{U} h_{s,u}^{(r)} \rangle = \frac{1}{U} \sum_{u=1}^{U} \langle \theta_{s,g}^{(r)} - \theta_{s,g}^*, h_{s,u}^{(r)} \rangle
$$

$$
= \frac{1}{U} \sum_{u=1}^{U} \sum_{e=0}^{E-1} \langle \theta_{s,g}^{(r)} - \theta_{s,g}^*, \nabla_{\theta_{s,u}^{(r,e)}} \mathcal{L}_u(w_{\text{LAG},u}^{(r,e)}) \rangle \tag{43}
$$

We have:
$$
\langle \theta_{s,g}^{(r)} - \theta_{s,g}^*, \nabla_{\theta_{s,u}^{(r,e)}} \mathcal{L}_u(w_{\text{LAG},u}^{(r,e)}) \rangle = \underbrace{\langle \theta_{s,g}^{(r)} - \theta_{s,g}^{(r,e)}, \nabla_{\theta_{s,u}^{(r,e)}} \mathcal{L}_u(w_{\text{LAG},u}^{(r,e)}) \rangle}_{Q_1^1} + \underbrace{\langle \theta_{s,g}^{(r,e)} - \theta_{s,g}^*, \nabla_{\theta_{s,u}^{(r,e)}} \mathcal{L}_u(w_{\text{LAG},u}^{(r,e)}) \rangle}_{Q_1^2}
$$
$$\tag{44}$$

Next, we need to consider two terms $Q_1$ and $Q_2$. Take $Q_1^1$ into consideration, due to the $L$-smooth in Assumption H.1, we have:

$$
Q_1^1 = \langle \theta_{s,g}^{(r)} - \theta_{s,g}^{(r,e)}, \nabla_{\theta_{s,u}^{(r,e)}} \mathcal{L}_u(w_{\text{LAG},u}^{(r,e)}) \rangle \geq \mathcal{L}_u(\theta_{s,g}^{(r)}) - \mathcal{L}_u(\theta_{s,g}^{(r,e)}) - \frac{L}{2} \|\theta_{s,g}^{(r)} - \theta_{s,g}^{(r,e)}\|^2. \tag{45}
$$

Take $Q_1^2$ into consideration, from Assumption H.2, we have:

$$
Q_1^2 = \langle \theta_{s,g}^{(r,e)} - \theta_{s,g}^*, \nabla_{\theta_{s,u}^{(r,e)}} \mathcal{L}_u(w_{\text{LAG},u}^{(r,e)}) \rangle \geq \mathcal{L}_u(\theta_{s,g}^{(r,e)}) - \mathcal{L}_u(\theta_{s,g}^*) + \frac{\mu}{2} \|\theta_{s,g}^{(r,e)} - \theta_{s,g}^*\|^2
$$
$$
\geq \mathcal{L}_u(\theta_{s,g}^{(r,e)}) - \mathcal{L}_u(\theta_{s,g}^*). \tag{46}
$$

Therefore, adding the above inequalities equation 45 and equation 46 together, we have:

$$
\langle \theta_{s,g}^{(r)} - \theta_{s,g}^*, \nabla_{\theta_{s,u}^{(r,e)}} \mathcal{L}_u(w_{\text{LAG},u}^{(r,e)}) \rangle \geq \mathcal{L}_u(\theta_{s,g}^{(r)}) - (\theta_{s,g}^*) - \frac{L}{2} \|\theta_{s,g}^{(r)} - \theta_{s,g}^{(r,e)}\|^2 \tag{47}
$$

Substituting equation 47 into equation 43, we have:

$$\begin{aligned}
Q_1 &= \frac{1}{U}\sum_{u=1}^{U}\sum_{e=0}^{E-1}\left[\langle\theta_{s,g}^{(r)}-\theta_{s,g}^{*},\nabla_{\theta_{s,u}^{(r,e)}}\mathcal{L}_u(w_{\text{LAG},u}^{(r,e)})\rangle\right] \\
&\geq \frac{1}{U}\sum_{u=1}^{U}\sum_{e=0}^{E-1}\left[\mathcal{L}_u(\theta_{s,g}^{(r)})-\mathcal{L}_u(\theta_{s,g}^{*})-\frac{L}{2}\|\theta_{s,g}^{(r)}-\theta_{s,g}^{(r,e)}\|^2\right] \\
&= E\left[\mathcal{L}_u(\theta_{s,g}^{(r)})-\mathcal{L}_u(\theta_{s,g}^{*})\right]-\frac{L}{2U}\sum_{u=1}^{U}\sum_{e=0}^{E-1}\|\theta_{s,g}^{(r)}-\theta_{s,g}^{(r,e)}\|^2 \qquad (48)
\end{aligned}$$

**Bounding $Q_2$:** We have:

$$Q_2 = \frac{1}{U}\sum_{u=1}^{U}\langle\theta_{p,u}^{(r)}-\theta_{p,g}^{*},h_{p,u}^{(r)}\rangle = \frac{1}{U}\sum_{u=1}^{U}\sum_{e=0}^{E-1}\langle\theta_{p,u}^{(r)}-\theta_{p,g}^{*},\nabla_{\theta_{p,u}^{(r,e)}}\mathcal{L}_u(w_{\text{LAG},u}^{(r,e)})\rangle \qquad (49)$$

We have:

$$\langle\theta_{p,u}^{(r)}-\theta_{p,g}^{*},\nabla_{\theta_{p,u}^{(r,e)}}\mathcal{L}_u(w_{\text{LAG},u}^{(r,e)})\rangle = \underbrace{\langle\theta_{p,u}^{(r)}-\theta_{p,u}^{(r,e)},\nabla_{\theta_{p,u}^{(r,e)}}\mathcal{L}_u(w_{\text{LAG},u}^{(r,e)})\rangle}_{Q_2^1} + \underbrace{\langle\theta_{p,u}^{(r,e)}-\theta_{p,g}^{*},\nabla_{\theta_{p,u}^{(r,e)}}\mathcal{L}_u(w_{\text{LAG},u}^{(r,e)})\rangle}_{Q_2^2}$$

$$\tag{50}$$

Take $Q_2^1$ into consideration, due to the $L$-smooth in Assumption H.1, we have:

$$Q_2^1 = \langle\theta_{p,u}^{(r)}-\theta_{p,u}^{(r,e)},\nabla_{\theta_{p,u}^{(r,e)}}\mathcal{L}_u(w_u^{(r,e)})\rangle \geq \mathcal{L}_u(\theta_{p,u}^{(r)})-\mathcal{L}_u(\theta_{p,u}^{(r,e)})-\frac{L}{2}\|\theta_{p,u}^{(r)}-\theta_{p,u}^{(r,e)}\|^2 \qquad (51)$$

Take $Q_2^2$ into consideration, from Assumption H.2, we have:

$$\begin{aligned}
Q_2^2 &= \langle\theta_{p,u}^{(r,e)}-\theta_{p,g}^{*},\nabla_{\theta_{p,u}^{(r,e)}}\mathcal{L}_u(w_u^{(r,e)})\rangle \geq \mathcal{L}_u(\theta_{p,u}^{(r,e)})-\mathcal{L}_u(\theta_{p,g}^{*})+\frac{\mu}{2}\|\theta_{p,u}^{(r,e)}-\theta_{p,g}^{*}\|^2 \\
&\geq \mathcal{L}_u(\theta_{p,u}^{(r,e)})-\mathcal{L}_u(\theta_{p,g}^{*}) \qquad (52)
\end{aligned}$$

Therefore, adding the above inequalities equation 51 and equation 52 together, we have:

$$\langle\theta_{p,u}^{(r)}-\theta_{p,g}^{*},\nabla_{\theta_{p,u}^{(r,e)}}\mathcal{L}_u(w_u^{(r,e)})\rangle \geq \mathcal{L}_u(\theta_{p,u}^{(r)})-\mathcal{L}_u(\theta_{p,g}^{*})-\frac{L}{2}\|\theta_{p,u}^{(r)}-\theta_{p,u}^{(r,e)}\|^2 \qquad (53)$$

Substituting equation 53 into equation 49, we have:

$$\begin{aligned}
Q_2 &= \frac{1}{U}\sum_{u=1}^{U}\sum_{e=0}^{E-1}\langle\theta_{p,u}^{(r)}-\theta_{p,g}^{*},\nabla_{\theta_{p,u}^{(r,e)}}\mathcal{L}_u(w_u^{(r,e)})\rangle \\
&\geq \frac{1}{U}\sum_{u=1}^{U}\left[\sum_{e=0}^{E-1}\mathcal{L}_u(\theta_{p,u}^{(r)})-\mathcal{L}_u(\theta_{p,g}^{*})-\frac{L}{2}\|\theta_{p,u}^{(r)}-\theta_{p,u}^{(r,e)}\|^2\right] \\
&= \frac{E}{U}\sum_{u=1}^{U}\left[\mathcal{L}_u(\theta_{p,u}^{(r)})-\mathcal{L}_u(\theta_{p,g}^{*})\right]-\frac{L}{2U}\sum_{u=1}^{U}\sum_{e=0}^{E-1}\|\theta_{p,u}^{(r)}-\theta_{p,u}^{(r,e)}\|^2 \qquad (54)
\end{aligned}$$

**Bounding $Q_3$:** We have:

$$\begin{aligned}
Q_3 &= \frac{1}{U}\sum_{u=1}^{U}\|h_{s,u}^{(r)}\|^2 = \frac{1}{U}\sum_{u=1}^{U}\left\|\sum_{e=0}^{E-1}\nabla_{\theta_{s,u}^{(r,e)}}\mathcal{L}_u(w_u^{(r,e)})\right\|^2 \overset{(a)}{\leq} \frac{E}{U}\sum_{u=1}^{U}\sum_{e=0}^{E-1}\left\|\nabla_{\theta_{s,u}^{(r,e)}}\mathcal{L}_u(w_u^{(r,e)})\right\|^2 \\
&\overset{(b)}{\leq} \frac{3EL^2}{U}\sum_{u=1}^{U}\sum_{e=0}^{E-1}\|\theta_{s,u}^{(r,e)}-\theta_{s,u}^{(r)}\|^2 + 6E^2L(\mathcal{L}_u(\theta_{s,g}^{(r)})-\mathcal{L}_u(\theta_{s,g}^{*})) + 3E^2\sigma_{s,*}^2. \qquad (55)
\end{aligned}$$

where (a) holds due to lemma H.5, and (b) holds due to lemma H.10.

**Bounding $Q_4$:** We have:

$$
\begin{aligned}
Q_4 &= \frac{1}{U} \sum_{u=1}^{U} \|h_{p,u}^{(r)}\|^2 = \frac{1}{U} \sum_{u=1}^{U} \Big\| \sum_{e=0}^{E-1} \nabla_{\theta_{s,u}^{(r,e)}} \mathcal{L}_u(w_u^{(r,e)}) \Big\|^2 \le \frac{E}{U} \sum_{u=1}^{U} \sum_{e=0}^{E-1} \Big\| \nabla_{\theta_{s,u}^{(r,e)}} \mathcal{L}_u(w_u^{(r,e)}) \Big\|^2 \\
&\le \frac{3E}{U} \sum_{u=1}^{U} \sum_{e=0}^{E-1} \| \nabla_{\theta_{p,u}^{(r,e)}} \mathcal{L}_u(w_u^{(r,e)}) - \nabla_{\theta_{p,u}^{(r,e)}} \mathcal{L}_u(w_{p,u}^{(r)}) \|^2 \\
&\quad + \frac{3E}{U} \sum_{u=1}^{U} \sum_{e=0}^{E-1} \| \nabla_{\theta_{p,u}^{(r)}} \mathcal{L}_u(w_{p,u}^{(r)}) - \nabla_{\theta_{p,g}^*} \mathcal{L}_u(w^*) \|^2 + \frac{3E}{U} \sum_{u=1}^{U} \sum_{e=0}^{E-1} \| \nabla_{\theta_{p,g}^*} \mathcal{L}_u(w^*) \|^2 \\
&\le \frac{3EL^2}{U} \sum_{u=1}^{U} \sum_{e=0}^{E-1} \|\theta_{p,u}^{(r,e)} - \theta_{p,u}^{(r)}\|^2 + \frac{6E^2 L}{U} \sum_{u=1}^{U} \Big( \mathcal{L}_u(\theta_{p,u}^{(r)}) - \mathcal{L}_u(\theta^*) \Big) + 3E^2 \sigma_{p,*}^2 \qquad (56)
\end{aligned}
$$

Combining equation 48, equation 54, equation 55, equation 56 together, we have:

$$
\begin{aligned}
&\mathbb{E}\Big[ \|w_{\text{LAG},g}^{(r+1)} - w^*\|^2 \Big] \\
&= \mathbb{E}\Big[ \|w_{\text{LAG},g}^{(r)} - w^*\|^2 \Big] - Q_1 + Q_3 - Q_2 + Q_4 \\
&= \mathbb{E}\Big[ \|w_{\text{LAG},g}^{(r)} - w^*\|^2 \Big] - 2\eta \underbrace{\Big\{ E\Big[ \mathcal{L}_u(\theta_{s,g}^{(r)}) - \mathcal{L}_u(\theta_{s,g}^*) \Big] - \frac{L}{2U} \sum_{u=1}^{U} \sum_{e=0}^{E-1} \|\theta_{s,g}^{(r)} - \theta_{s,g}^{(r,e)}\|^2 \Big\}}_{Q_1} \\
&\quad + \eta^2 \underbrace{\Big\{ \frac{3EL^2}{U} \sum_{u=1}^{U} \sum_{e=0}^{E-1} \|\theta_{s,u}^{(r,e)} - \theta_{s,u}^{(r)}\|^2 + 6E^2 L(\mathcal{L}_u(\theta_{s,g}^{(r)}) - \mathcal{L}_u(\theta_{s,g}^*)) + 3E^2 \sigma_{s,*}^2 \Big\}}_{Q_3} \\
&\quad - 2\eta \underbrace{\Big\{ \frac{E}{U} \sum_{u=1}^{U} \Big[ \mathcal{L}_u(\theta_{p,u}^{(r)}) - \mathcal{L}_u(\theta_{p,g}^*) \Big] - \frac{L}{2U} \sum_{u=1}^{U} \sum_{e=0}^{E-1} \|\theta_{p,g}^{(r)} - \theta_{p,g}^{(r,e)}\|^2 \Big\}}_{Q_2} \\
&\quad + \eta^2 \underbrace{\Big\{ \frac{3EL^2}{U} \sum_{u=1}^{U} \sum_{e=0}^{E-1} \|\theta_{p,u}^{(r,e)} - \theta_{p,u}^{(r)}\|^2 + \frac{6E^2 L}{U} \sum_{u=1}^{U} \Big( \mathcal{L}_u(\theta_{p,u}^{(r)}) - \mathcal{L}_u(\theta^*) \Big) + 3E^2 \sigma_{p,*}^2 \Big\}}_{Q_4} \\
&= \mathbb{E}\Big[ \|w_{\text{LAG},g}^{(r)} - w^*\|^2 \Big] + \frac{\eta^2 L}{U} \sum_{u=1}^{U} \sum_{e=0}^{E-1} \Big\{ \|\theta_{s,g}^{(r)} - \theta_{s,g}^{(r,e)}\|^2 + \|\theta_{p,g}^{(r)} - \theta_{p,g}^{(r,e)}\|^2 \Big\} \\
&\quad + \frac{3E\eta^2 L^2}{U} \sum_{u=1}^{U} \sum_{e=0}^{E-1} \Big\{ \|\theta_{s,u}^{(r,e)} - \theta_{s,u}^{(r)}\|^2 + \|\theta_{p,u}^{(r,e)} - \theta_{p,u}^{(r)}\|^2 \Big\} \\
&\quad - 2\frac{\eta E}{U} \sum_{u=1}^{U} \Big\{ \Big[ \mathcal{L}_u(\theta_{p,u}^{(r)}) - \mathcal{L}_u(\theta_{p,g}^*) \Big] + \Big[ \mathcal{L}_u(\theta_{s,g}^{(r)}) - \mathcal{L}_u(\theta_{s,g}^*) \Big] \Big\} \\
&\quad + \frac{6E^2 \eta^2 L}{U} \sum_{u=1}^{U} \Big\{ \Big( \mathcal{L}_u(\theta_{s,g}^{(r)}) - \mathcal{L}_u(\theta_{s,g}^*) \Big) + \Big( \mathcal{L}_u(\theta_{p,u}^{(r)}) - \mathcal{L}_u(\theta^*) \Big) \Big\} + 3E^2 \eta^2 \sigma_*^2. \qquad (57)
\end{aligned}
$$

Combine together, we have:

$$\mathbb{E}\Big[\|w_{\text{LAG},g}^{(r+1)} - w^*\|^2\Big]$$

$$= \mathbb{E}\Big[\|w_{\text{LAG},g}^{(r)} - w^*\|^2\Big] + \frac{\eta L}{U}\sum_{u=1}^{U}\sum_{e=0}^{E-1}\Big\{\|\theta_{s,g}^{(r)} - \theta_{s,g}^{(r,e)}\|^2 + \|\theta_{p,g}^{(r)} - \theta_{p,g}^{(r,e)}\|^2\Big\}$$

$$+ \frac{3E\eta^2 L^2}{U}\sum_{u=1}^{U}\sum_{e=0}^{E-1}\Big\{\|\theta_{s,u}^{(r,e)} - \theta_{s,u}^{(r)}\|^2 + \|\theta_{p,u}^{(r,e)} - \theta_{p,u}^{(r)}\|^2\Big\}$$

$$- 2\frac{\eta E}{U}\sum_{u=1}^{U}\Big[\mathcal{L}_u(w_{\text{LAG},u}^{(r)}) - \mathcal{L}_u(w^*)\Big] + \frac{6E^2\eta^2 L}{U}\sum_{u=1}^{U}\Big(\mathcal{L}_u(w_{\text{LAG},u}^{(r)}) - \mathcal{L}_u(w^*)\Big) + 3E^2\eta^2\sigma_*^2$$

$$= \mathbb{E}\Big[\|w_{\text{LAG},g}^{(r)} - w^*\|^2\Big] + (3E\eta^2 L^2 + \eta L)\frac{1}{U}\sum_{u=1}^{U}\sum_{e=0}^{E-1}\|w_{\text{LAG},u}^{(r,e)} - w_{\text{LAG},u}^{(r)}\|^2$$

$$- 2\frac{\eta E}{U}(1 - 3E\eta L)\sum_{u=1}^{U}\Big[\mathcal{L}_u(w_{\text{LAG},u}^{(r)}) - \mathcal{L}_u(w^*)\Big] + 3E^2\eta^2\sigma_*^2$$

$$\overset{(a)}{\leq} \mathbb{E}\Big[\|w_{\text{LAG},g}^{(r)} - w^*\|^2\Big] + \frac{2\eta L}{U}\sum_{u=1}^{U}\sum_{e=0}^{E-1}\|w_{\text{LAG},u}^{(r,e)} - w_{\text{LAG},u}^{(r)}\|^2$$

$$- \frac{\eta E}{U}\sum_{u=1}^{U}\Big[\mathcal{L}_u(w_{\text{LAG},u}^{(r)}) - \mathcal{L}_u(w^*)\Big] + 3E^2\eta^2\sigma_*^2$$

$$= \mathbb{E}\Big[\|w_{\text{LAG},g}^{(r)} - w^*\|^2\Big] + 2\eta L\Big\{12E^2(E-1)\eta^2 L\Big(\mathcal{L}(w^{(r)} - \mathcal{L}(w^*)\Big)$$

$$- 12E^2(E-1)\eta^3\frac{L}{U^2}\sum_{u=1}^{U}\sum_{v=1}^{U}\sum_{l\in\mathbb{L}_p}\|h_{l,u}^{(r)}\|\Big(\|h_{l,u}^{(r)}\| - \cos\Phi_{u,v}^{(l)}\|h_{l,v}^{(r)}\|\Big) + 6\eta^2 E^2(E-1)\sigma_*^2\Big\}$$

$$- \frac{\eta E}{U}\Big\{\sum_{u=1}^{U}\Big(\mathcal{L}(w_g^{(r)} - \mathcal{L}(w^*)\Big) - \frac{\eta}{U}\sum_{u=1}^{U}\sum_{v=1}^{U}\sum_{l\in\mathbb{L}_p}\|h_{l,u}^{(r)}\|\Big(\|h_{l,u}^{(r)}\| - \cos\Phi_{u,v}^{(l)}\|h_{l,v}^{(r)}\|\Big) + 3E^2\eta^2\sigma_*^2.$$

(58)

Rearrange terms, we have:

$$\mathbb{E}\Big[\|w_{\text{LAG},g}^{(r+1)} - w^*\|^2\Big]$$

$$= \mathbb{E}\Big[\|w_{\text{LAG},g}^{(r)} - w^*\|^2\Big] + \Big(24E^2(E-1)\eta^3 L^2 - \eta E\Big)\frac{1}{U}\sum_{u=1}^{U}\Big(\mathcal{L}(w_{\text{LAG},u}^{(r)} - \mathcal{L}(w^*)\Big)$$

$$- \frac{1}{U^2}\Big(24E^2(E-1)\eta^4 L^2 - \eta^2 E\Big)\sum_{u=1}^{U}\sum_{v=1}^{U}\sum_{l\in\mathbb{L}_p}\|h_{l,u}^{(r)}\|\Big(\|h_{l,u}^{(r)}\| - \cos\Phi_{u,v}^{(l)}\|h_{l,v}^{(r)}\|\Big)$$

$$+ 12\eta^3 E^2(E-1)L\sigma_*^2 + 3E^2\eta^2\sigma_*^2.$$

(59)

Here, (a) due to the assumption that $\eta < \frac{1}{6EL}$. For local epoch $E > 1$, we have $E \geq \sqrt{3}$, which means that $\frac{1}{2\sqrt{6}E^2 L} < \eta < \frac{1}{6EL}$. Thus, we have $\frac{1}{U^2}\Big(24E^2(E-1)\eta^4 L^2 - \eta^2 E\Big) > 0$, $\forall \eta > \frac{1}{2\sqrt{6}E^2 L}$ (the proof is provided in Appendix I). Therefore, we can rewrite the term as:

$$\|w_{\text{LAG},g}^{(r+1)} - w^*\|^2 \leq \|w_{\text{LAG},g}^{(r)} - w^*\|^2 - \frac{\eta E}{3}\Big(\mathcal{L}(w_g^{(r)} - \mathcal{L}(w^*)\Big) + \Big[3E\eta^2 + 12\eta^3 E^2(E-1)L\Big]\sigma_*^2$$

$$- \frac{24}{U^2}A\sum_{u=1}^{U}\sum_{v=1}^{U}\sum_{l\in\mathbb{L}_p}\|h_{p,u}^{(r)}\|(\|h_{p,u}^{(r)}\| - \cos\Phi_{u,v}^{(l)}\|h_{p,v}^{(r)}\|),$$

(60)

where $A = \Big(24E^2(E-1)\eta^4 L^2 - \eta^2 E\Big) > 0$, thus, the algorithm always give a consistent improvement gap to the global learning performance.

Due to Lemma H.13 and Lemma H.11. Combine together, we have:

$$\|w_{\text{LAG},g}^{(r+1)} - w^*\|^2 \leq \|w_{\text{LAG},g}^{(r)} - w^*\|^2 - \frac{\eta E}{3}\Big(\mathcal{L}(w_g^{(r)} - \mathcal{L}(w^*))\Big) + \Big[3E\eta^2 + 12\eta^3 E^2(E-1)L\Big]\sigma_*^2$$
$$- \mathcal{O}\Big(\sum_{u=1}^{U}\sum_{v=1}^{U}\sum_{l\in\mathbb{L}_p} \|h_{l,u}^{(r)}\|\Big(\|h_{l,u}^{(r)}\| - \cos\Phi_{u,v}^{(l)}\|h_{l,v}^{(r)}\|\Big)\Big). \tag{61}$$

Moreover, we have $\eta > \frac{1}{2\sqrt{6}E^2L} > \frac{1}{2\sqrt{6}EL}$ (as $E > 1$). Therefore, we have Rearranging terms and averaging over all rounds, we have:

$$\|w_{\text{LAG},g}^{(r+1)} - w^*\|^2 \leq \|w_{\text{LAG},g}^{(0)} - w^*\|^2 - \sum_{r=0}^{R-1}\frac{\eta E}{3}\Big(\mathcal{L}(w_g^{(r)} - \mathcal{L}(w^*))\Big) + \sum_{r=0}^{R-1}\Big[3E\eta^2 + 12\eta^3 E^2(E-1)L\Big]\sigma_*^2$$
$$- \mathcal{O}\Big(\sum_{u=1}^{U}\sum_{v=1}^{U}\sum_{l\in\mathbb{L}_p} \|h_{l,u}^{(r)}\|\Big(\|h_{l,u}^{(r)}\| - \cos\Phi_{u,v}^{(l)}\|h_{l,v}^{(r)}\|\Big)\Big). \tag{62}$$

$$\mathcal{L}(w_g^{(r)}) - \mathcal{L}(w^*) \leq \frac{3}{R\eta E}\|w_{\text{LAG},g}^{(0)} - w^*\|^2 - \frac{3}{R\eta E}\|w_{\text{LAG},g}^{(r+1)} - w^*\|^2 + \frac{3}{\eta E}\Big[3E\eta^2 + 12\eta^3 E^2(E-1)L\Big]\sigma_*^2$$
$$- \mathcal{O}\Big(\sum_{u=1}^{U}\sum_{v=1}^{U}\sum_{l\in\mathbb{L}_p} \|h_{l,u}^{(r)}\|\Big(\|h_{l,u}^{(r)}\| - \cos\Phi_{u,v}^{(l)}\|h_{l,v}^{(r)}\|\Big)\Big)$$
$$\leq \frac{3\|w_{\text{LAG},g}^{(0)} - w^*\|^2}{R\eta E} + 9\eta\sigma_*^2 + 36\eta^2 E(E-1)L\sigma_*^2$$
$$- \mathcal{O}\Big(\sum_{u=1}^{U}\sum_{v=1}^{U}\sum_{l\in\mathbb{L}_p} \|h_{l,u}^{(r)}\|\Big(\|h_{l,u}^{(r)}\| - \cos\Phi_{u,v}^{(l)}\|h_{l,v}^{(r)}\|\Big)\Big). \tag{63}$$

As we have aggregated the layer-wise personalized model and global aggregate model back into $w_{\text{LAG},g}^{(r)}$, we can have:

$$\mathcal{L}(w_g^{(R)}) - \mathcal{L}(w^*) \leq \mathcal{O}\Big(\frac{\|w_g^{(0)} - w^*\|^2}{R\eta E}\Big) + \mathcal{O}\Big(\eta\sigma_*^2\Big) + \mathcal{O}\Big(\eta^2 E(E-1)L\sigma_*^2\Big)$$
$$- \mathcal{O}\Big(\sum_{u=1}^{U}\sum_{v=1}^{U}\sum_{l\in\mathbb{L}_p} \|h_{l,u}^{(r)}\|\Big(\|h_{l,u}^{(r)}\| - \cos\Phi_{u,v}^{(l)}\|h_{l,v}^{(r)}\|\Big)\Big). \tag{64}$$

This completes the proof.

# I   PROOF ON CONSISTENT IMPROVEMENT GAP

Consider a second-degree polynomial function given by $Y = AX^2 - BX$, where $A > 0$. Consequently, we ensure $Y > 0$ under the condition $X \in (-\infty, 0] \cup [B/A; +\infty)$.

Next, we consider the function $Y = E^2(E-1)\eta^4 L^2 - \eta^2 E$, and take $\eta^2 = X$. This yields:

$$Y = 24E^2(E-1)L^2 X^2 - EX. \tag{65}$$

Given that $E^2(E-1)L^2 \geq 0$ for all $E, L$. To ensure $Y > 0$, we have to satisfy the following condition:

$$X \geq \frac{E}{24E^2(E-1)L^2} = \frac{1}{24E(E-1)L^2}. \tag{66}$$

Substituting $X$ back to $\eta^2$, then we have

$$\eta^2 \geq \frac{1}{24E(E-1)L^2}. \tag{67}$$

Accordingly, we

$$\eta \geq \frac{1}{2\sqrt{6}EL}. \tag{68}$$

Given that $\frac{1}{2\sqrt{6}E^2 L} < \eta < \frac{1}{6EL}$, then, $\eta \geq \frac{1}{2\sqrt{6}E^2 L} > \frac{1}{2\sqrt{6}EL}$, which holds that $\eta$ always satisfy the condition to have $Y > 0$.

# J   ANALYSIS ON COMPUTATIONAL COMPLEXITY OF ON-SERVER GRADIENT ANALYSIS

We consider the time complexity of the feed-forward process of an AI model. Consider a model with $L$ hidden layers, denoted as $\{w_1, w_2, \ldots, w_{d_l}\}$ representing the weights of each layer. For simple approximation, we assume that all layers have same number of parameters, i.e., $d_1 = d_2 = \ldots = d_L = D$.

We consider the feed-forwarding task of an AI model with a batch size of $B$.

To go through $L$ layers, each layers have $D$ parameters, we have to do $y = W_{LD} \times x$. Therefore, we have the feed-forward process for a mini-batch with batch size $B$ has time complexity $\mathcal{O}(L \times D \times B \times S)$, where $S$ is the data size.

Second, according to the personalized layers selection, the time complexity is around $\mathcal{O}(U \times (U - 1)/2 \times D \times L)$. If we consider a large number of users $U = 1000$, and due to the large DNN, we use ImageNet as evaluations, where the average image dimensionality is $3 \times 469 \times 387 = 544509$. Therefore, we have the two complexity two process as follows:

- One iteration of feed-forward in a DNN with only one sample: $\mathcal{O}(L \times D \times 1 \times 544509)$.
- Personalized layers selection: $\mathcal{O}(500 \times 999 \times D \times L)$.

It is obvious that the personalized layers selection is approximately same with the time consumption of 1 iteration of feed-forward in DNN.

Finally, we survey and found that the inference time in Large DNN Model is trivial ($\ll$ 1s/sample). The detailed results can be found in (Liu et al., 2022, Table 1).

## K    DETAILED ALGORITHMS

---

**Algorithm 1:** Layer-wise Gradient Analysis supported pFL

---

**Require :** Initialize users' weights, number of users $U$.

1 **while** *not converge at round $r$* **do**
2     **On-user Training**
3     **for** *each user $u \in \{1, 2, \ldots, U\}$* **do**
4         **for** *epoch $e \in E$* **do**
5             Sample mini-batch $\zeta$ from local data $\mathcal{D}_u$
6             Calculate gradient $g_u^{(r,e)} = \nabla \mathcal{L}(w_u^{(r,e)}, \zeta)$
7             Update user's model $w_u^{(r,e+1)} = w_u^{(r,e)} - \eta g_u^{(r,e)}$.
8         **end for**
9         Upload user's model $w_u^{(r,E)}$ to server.
10     **end for**
11     **On-server Training**
12     Find personalized layers $\mathbb{L}_p$ from users' model parameters via Algorithm 2.
13     Aggregate users model by $w_g^{(r+1)} = \frac{1}{U} \sum_{u=1}^{U} w_u^{(r,E)}$ and save to storage to calculate the gradient for the next round.
14     **Broadcast aggregated models to local users**
15     **for** *each layer $l \in \mathbb{L}_p$* **do**
16         Update the local model of user $u$ as follows:
17

$$\theta_{l,u}^{(r+1)} = \begin{cases} \theta_{l,u}^{(r,E)}, & \text{if } l \in \mathbb{L}_p, \\ \frac{1}{U} \sum_{u=1}^{U} \theta_{l,u}^{(r,E)}, & \text{otherwise.} \end{cases}$$

18     **end for**
19 **end while**

---

---

**Algorithm 2:** Layer-wise Gradient Divergence Analysis

---

**Require :** users' models $w_1^{(r,E)}, w_2^{(r,E)}, \ldots, w_U^{(r,E)}$ collected in round $r$, previous models $w_1^{(r)}, w_2^{(r)}, \ldots, w_U^{(r)}$ stored from last round, threshold $\xi$, number of personalized layers $k$.

1 **On-server Training**
2 **for** *all layers $l \in \{1, 2, \ldots, L\}$* **do**
3     Calculate $h_{l,u}^{(r)}$ from $w_u^{(r,E)}$, $w_u^{(r)}$ via Eq. equation 2.
4     Make pairs $\{h_{l,u}^{(r)}, h_{l,v}^{(r)}\}, \forall u, v \in U, u \neq v$.
5     Obtain cosine according to each $\cos \phi_l^{(r)}(u, v)$.
6     Calculate $GC_\xi(l)$ according to Eq. equation 3.
7 **end for**
8 Assign layer with top $k$ highest $GC_\xi(l)$ to $\mathbb{L}_p$.
9 **Return:** $\mathbb{L}_p$

---

