# OpenReview forum: "TOWARDS LAYER-WISE PERSONALIZED FEDERATED LEARNING: ADAPTIVE LAYER DISENTANGLEMENT VIA CONFLICTING GRADIENTS"
_ICLR.cc/2025/Conference — ICLR 2025 Conference Withdrawn Submission_

### Official Review · Reviewer_1fDg · 2024-10-31

**Soundness:** 3
**Presentation:** 1
**Contribution:** 2
**Rating:** 5
**Confidence:** 4

**Summary:**

This paper presents a novel layer-wise aggregation method for personalized federated learning, named FedLAG. By analyzing gradient conflicts, FedLAG dynamically determines which layers should undergo global aggregation (GAL) and which should be personalized (PL) to mitigate the negative impact of data heterogeneity on federated learning. FedLAG adapts to the personalized needs of the data, reducing the reliance on prior knowledge that traditional methods require, thereby enhancing model convergence and adaptability. Experimental results indicate that FedLAG outperforms existing baseline methods across multiple datasets.

**Strengths:**

1)Innovation: This paper addresses a key challenge in personalized federated learning, specifically, how to implement layer-wise model partitioning under data heterogeneity, and introduces an innovative method based on gradient conflict analysis.
2)Theoretical Rigor: The paper provides a theoretical analysis demonstrating the convergence advantages of FedLAG and includes mathematical proofs to validate the effectiveness of its gradient conflict-based layer separation.
3)Comprehensive Experimental Design: The experiments encompass various datasets of data heterogeneity, with results affirming FedLAG's strengths in both accuracy and convergence speed.

**Weaknesses:**

1)Gradient Conflict Concept Insufficiently Explained: Although the paper defines gradient conflict, it lacks examples or illustrations to show how gradient conflicts are detected and quantified in practice, especially in the context of layer separation. Additionally, the choice of the gradient conflict threshold , as well as its impact on model performance, is not clearly explained. The paper merely describes conflict-based layer separation without providing an intuitive example. Supplementary examples or visual aids are recommended to enhance reader comprehension.
2)Inadequate Ablation Studies: While the paper compares the performance of fixed layers, it does not explain why certain layers are more suitable for personalization than others. This section lacks theoretical support or empirical evidence, rendering the rationale for layer selection ambiguous. Further experiments or analyses could substantiate the impact of layer choice on model performance, strengthening the logic behind model design.
3)Inaccurate Terminology: The paper states that “existing methods require extensive fine-tuning to determine which layers should be used for global aggregation and personalization” characterizing a limitation of existing methods. However, the term “fine-tuning” is inappropriate in this context. References [1] and [2] emphasize the role of prior knowledge in layer separation, and clarifying this terminology would help prevent potential misunderstandings.
4)Ambiguous Notation: Several notations in Section 2 are unclear, such as the definition of global communication rounds. This paper uses “each round” to describe this symbol, which may confuse readers regarding whether it refers to global communication rounds or local iteration rounds. Furthermore, in the “Layer Disentanglement” section, the complex and nested notation complicates the reader's understanding of the hierarchical structure. A clearer description of the relationships among symbols is recommended, along with a well-organized explanation of global and personalized layers. Visual aids, such as diagrams, could also serve as useful tools to help readers intuitively grasp the layered model structure.
5)Insufficient Discussion on Limitations and Future Directions: The paper does not discuss the limitations of the proposed approach or provide insights into future research directions in the conclusion. It is recommended to analyze the limitations, such as the uncertain performance on more complex types of heterogeneous data (e.g., feature distribution skewed datasets) or potential applications in NLP and time-series analysis. Future research directions could also include exploring the method on multi-type heterogeneous datasets to enhance generalizability.
6)Lack of In-depth Analysis of Experimental Results: Table 1 and Figures 6 and 7 demonstrate FedLAG's performance across various datasets, but the analysis of performance differences is not sufficiently in-depth. For instance, FedLAG's substantial improvement over other methods on CIFAR-10 and CIFAR-100 is not explained in terms of potential causes or contributing factors. A detailed analysis of these phenomena would increase the paper's credibility and persuasive power.
7)Insufficient Validation of Gradient Conflict Distribution: Although a 2D toy dataset experiment was conducted in Section 3, there is insufficient evidence to demonstrate the consistency of gradient conflict distribution across different datasets.
[1] Shen Y, Zhou Y, Yu L. Cd2-pfed: Cyclic distillation-guided channel decoupling for model personalization in federated learning[C]//Proceedings of the IEEE/CVF Conference on Computer Vision and Pattern Recognition. 2022: 10041-10050.
[2] Wu X, Liu X, Niu J, et al. Bold but cautious: Unlocking the potential of personalized federated learning through cautiously aggressive collaboration[C]//Proceedings of the IEEE/CVF International Conference on Computer Vision. 2023: 19375-19384.

**Questions:**

See the weakness.

---

> ### Author Response · Authors · 2024-11-21
> **Response to Reviewer 1fDg 's Weaknesses**
>
> > **Weakness 1:* Gradient Conflict Concept Insufficiently Explained: Although the paper defines gradient conflict, it lacks examples or illustrations to show how gradient conflicts are detected and quantified in practice, especially in the context of layer separation. Additionally, the choice of the gradient conflict threshold , as well as its impact on model performance, is not clearly explained. ...
>
>   In terms of visualization of gradient conflicts, we have demonstrated in Fig. 1 and empirically demonstrate an illustrative toy task in Fig. 2(a).
>
> > **Weakness 2:** While the paper compares the performance of fixed layers, it does not explain why certain layers are more suitable for personalization than others. This section lacks theoretical support or empirical evidence, rendering the rationale for layer selection ambiguous. Further experiments or analyses could substantiate the impact of layer choice on model performance, strengthening the logic behind model design.
>
>   In Section 5.1, we have demonstrated the impact of choosing layers with angles >90 degree to the general improvement of the FL system. We believe that our theoretical analysis has supported the rationale why choosing certain layer with angle > 90 degree is better than other layers.
>
> > **Weakness 3:** The paper states that “existing methods require extensive fine-tuning to determine which layers should be used for global aggregation and personalization” characterizing a limitation of existing methods. However, the term “fine-tuning” is inappropriate in this context. References [1] and [2] emphasize the role of prior knowledge in layer separation, and clarifying this terminology would help prevent potential misunderstandings.
>
>   We agree with the inappropriate “fine-tuning” in this context. We have revised the word to “extensive efforts for choosing appropriate hyper-parameters”.
>
> > **Weakness 4:** Several notations in Section 2 are unclear, such as the definition of global communication rounds. This paper uses “each round” to describe this symbol, which may confuse readers regarding whether it refers to global communication rounds or local iteration rounds. Furthermore, in the “Layer Disentanglement” section, the complex and nested notation complicates the reader's understanding of the hierarchical structure. ...
>
>   In our paper, we denote the global communication round and local epoch. As a consequence we think that the writing is consistent.
>
>   We apologize the Reviewer for the complex notations. However, in our paper, we focus on the layer disentanglement. As a consequence, a large number of notations needed to be defined for the convenience in showing the method and theoretical analysis comprehensively. We also provided the visualization in the Figure 3 which show the explanation of the system architecture along with the global and personalized layers. We would appreciate if the Reviewer can provide a more detailed advice, thus, we can try our best to satisfy the Reviewer.
>
> > **Weakness 5:** The paper does not discuss the limitations of the proposed approach or provide insights into future research directions in the conclusion. It is recommended to analyze the limitations, such as the uncertain performance on more complex types of heterogeneous data (e.g., feature distribution skewed datasets) or potential applications in NLP and time-series analysis. Future research directions could also include exploring the method on multi-type heterogeneous datasets to enhance generalizability.
>
>   We agree with the reviewer that providing discussion on limitations and future direction would improve the paper. We have revised the manuscript accordingly.
>
> > **Weakness 6:** Table 1 and Figures 6 and 7 demonstrate FedLAG's performance across various datasets, but the analysis of performance differences is not sufficiently in-depth. For instance, FedLAG's substantial improvement over other methods on CIFAR-10 and CIFAR-100 is not explained in terms of potential causes or contributing factors. A detailed analysis of these phenomena would increase the paper's credibility and persuasive power.
>
>   We agree with the reviewer’s suggestion, we have revised the manuscript accordingly to improve the paper’s credibility and persuasive power.
>
> > **Weakness 7:** Although a 2D toy dataset experiment was conducted in Section 3, there is insufficient evidence to demonstrate the consistency of gradient conflict distribution across different datasets.
>
>   We agree that 2D toy dataset is insufficient for the gradient conflict distributions among different layers and should be evaluated across different datasets. However, we believe that the 2D toy dataset can give intuitions of how can gradient conflicts affect to the learning progress of AI models.
>
>   From the perspectives of the gradient conflict distribution across different datasets, we have also conducted the evaluations on MNIST, and demonstrated in Appendix F.4

---

> > ### Comment · Reviewer_1fDg · 2024-11-27
> > **Thank you for the detailed response. The response has partially solved my concerns, I will keep my rating.**
> >
> > Thank you for the detailed response. The response has partially solved my concerns, I will keep my rating.

---

### Official Review · Reviewer_2xoi · 2024-11-01

**Soundness:** 3
**Presentation:** 3
**Contribution:** 2
**Rating:** 6
**Confidence:** 4

**Summary:**

This paper proposes a novel layer-wise personalized federated learning (PFL) method called FedLAG, which uses the divergence of client gradients as a criterion for selecting personalized layers. Experimental results demonstrate the effectiveness of this approach.

**Strengths:**

1. The paper is well-written, and the main claim is easy to understand.
2. Using gradient divergence as a criterion for selecting personalized layers seems reasonable.
3. Extensive experiments demonstrate the effectiveness of FedLAG.

**Weaknesses:**

1. Although gradient conflict is easy to understand, how gradient alignment specifically influences collaboration effectiveness in federated learning remains unclear. For instance, how large of an angle actually affects performance? Is an angle greater than 90 degrees necessarily detrimental? The paper primarily tunes the hyperparameter  \xi  to identify a threshold, but exploring the relationship between angle and performance further, along with providing a guideline for adjusting  \xi , would greatly enhance the quality of the work.
2. From the results in Tables 1 and 7, the performance improvement of FedLAG over the SOTA methods appears limited, with gains of less than 1%. Moreover, existing methods can already decouple the learning of personalized and generic knowledge in each parameter [1], I suggest that the authors compare FedLAG to this method.
3. As shown in Table 3, FedLAG appears highly sensitive to the hyperparameter  \xi . Combined with the results in Table 1, FedLAG requires very careful tuning of  \xi  to achieve performance superior to SOTA, which limits its usability.

[1] Decoupling General and Personalized Knowledge in Federated Learning via Additive and Low-Rank Decomposition, ACM MM 2024

**Questions:**

Section 3:

1. In Question 3.2, the authors state that “the density of conflicting gradients among clients does not progressively increase from the input layer to the output layer.” However, from Fig. 2(b), it appears that deeper layers have higher conflict scores. Could the authors clarify this discrepancy? I suggest that the authors provide a quantitative analysis of the trend in conflict scores across layers, rather than relying on a visual inspection of the figure.
2. In Section 4.1, why is FedLAG considered communication-efficient?

Section 7:

1. In 7.1.1, the authors mention “with improvements ranging from an average of 5–7% to 15%.” How was this calculated? From Table 1, it seems that FedLAG does not exceed the performance of the best method by more than 1% in most cases.
2. In 7.2.2, what would happen if  K  layers were randomly selected instead?
3. Which layers does FedLAG typically select for personalization?

---

> ### Author Response · Authors · 2024-11-21
> **Response to Reviewer 2xoi Weaknesses**
>
> > **Weakness 1:** Although gradient conflict is easy to understand, how gradient alignment specifically influences collaboration effectiveness in federated learning remains unclear. For instance, how large of an angle actually affects performance? Is an angle greater than 90 degrees necessarily detrimental? The paper primarily tunes the hyperparameter \xi to identify a threshold, but exploring the relationship between angle and performance further, along with providing a guideline for adjusting \xi , would greatly enhance the quality of the work.
>
>   The angle greater than 90 degree is necessarily detrimental. When two gradients form an obtuse angle, the gradient update of one task will cause negative transfer to another task [R1]. This phenomenon can be explained in terms of geometry. Specifically, when the angle of two gradients $g_1, g_2$ is obtuse, a specific gradient $g_2$ can be decomposed into two components $g^{\top}_2, g^{\parallel}_2$. $g^{\top}_2$ is orthogonal to $g_1$, thus, only focuses on helping $g_1$ to find an optimal trajectory.  $g^{\parallel}_2$ is the anti-parallel vector of $g_1$, thus, affects badly to the gradient progress of $g_1$.
>
>   Motivated from the aforementioned gradient conflicts in multi-task learning, the invariant gradient direction states that, if the two gradients hold an angle less than $90$ degree, the two gradients will progress towards a directions, which holds good performance on both domains. As a consequence, we can achieve domain-invariant representation with gradient direction alignment.
>
>   [R1] Adrian Javaloy et al., RotoGrad, Gradient Homogenization in Multi-task Learning, ICLR 2022.
>
> > **Weakness 2:** From the results in Tables 1 and 7, the performance improvement of FedLAG over the SOTA methods appears limited, with gains of less than 1%. Moreover, existing methods can already decouple the learning of personalized and generic knowledge in each parameter [1], I suggest that the authors compare FedLAG to this method.
>
>   The observed performance gain, which is less than 1%, primarily occurs in simpler datasets such as MNIST and EMNIST. This is likely due to the simplicity of these datasets, leading to comparable performance across different baseline models. In contrast, for more challenging datasets, such as CIFAR-10 and CIFAR-100, the improvement is notably more substantial.

---

> ### Author Response · Authors · 2024-11-21
> **Response to Reviewer 2xoi's Questions**
>
> > **Question 1:** In Question 3.2, the authors state that “the density of conflicting gradients among clients does not progressively increase from the input layer to the output layer.” However, from Fig. 2(b), it appears that deeper layers have higher conflict scores. Could the authors clarify this discrepancy? I suggest that the authors provide a quantitative analysis of the trend in conflict scores across layers, rather than relying on a visual inspection of the figure.
>
>   Although the deeper layers tend to have higher conflict scores, the conflict score is not compulsory distributed in an increasing order. For instance,
>
>   * In IID concept, the conflict scores are distributed among the layers with alternating high and low values.
>   * In non-IID concept, although the conflict scores are distributed with high scores at the very last layers, the score at the very last layers tend to reduce after 200 training rounds, and become approximately similar to that of layers 10-12.
>
> > **Question 2:** In Section 4.1, why is FedLAG considered communication-efficient?
>
>   In our work, we reuse the model weights to compute the gradients for the layer-wise analysis. As a consequence, FedLAG does not require any further data transmission to the server.
>
> > **Question 3:** In 7.1.1, the authors mention “with improvements ranging from an average of 5–7% to 15%.” How was this calculated? From Table 1, it seems that FedLAG does not exceed the performance of the best method by more than 1% in most cases.
>
>   We made a rough estimation and comparing FedLAG with the second best and worst performance in one dataset and settings. For example, in CIFAR100 with non-IID settings, FedLAG achieves 33% in terms of accuracy while FedAVG  only achieves 16% (more than 15%) and averagely the results is from 24-32%, which is around 5-7%. The observed performance gain, which is less than 1%, primarily occurs in simpler datasets such as MNIST and EMNIST. This is likely due to the simplicity of these datasets, leading to comparable performance across different baseline models.
>
> > **Question 5:** In 7.2.2, what would happen if K layers were randomly selected instead?
>
> We believe that randomly selected $K$ layers does not have an appropriate rationale supported behind them. As a consequence, we did not conduct the experiments with randomly K selected. Besides, we want to note that, randomly select $K$ layers every rounds may leads to the selection of majority of layers to be excluded for the averaging, which reduce the generalization of the FL system.
>
> > **Question 6:** Which layers does FedLAG typically select for personalization?
>
> As shown in Figure 2.b, it can be observed that in typical Non-IID data splits, the deeper layers tend to exhibit higher levels of conflict and are therefore more frequently selected for personalization. However, as we answered in Question 1, the very last layers are not distributed in a constantly increasing way in gradient conflict scores. Thus, If we choose an appropriate top $K$ value, the personalized layers are not all set at the very last layers.

---

> > ### Comment · Reviewer_2xoi · 2024-11-25
> >
> > 1. The authors claimed that “The angle greater than 90 degrees is necessarily detrimental.” However, as shown in Table 3, the optimal $\xi$ is -0.1, which indicates that in some cases, an angle greater than 90 degrees can still be beneficial; otherwise, $\xi=0$ should have been the optimal value. While I do not deny the authors’ statement that $g_2^{||}$ is harmful, I also believe that $g_2^{T}$ can still contribute positively to $g_1$. Therefore, I find the authors’ response insufficient to address my concern.
> > 2. Upon re-examining the table 1, I observed that the improvements of FedLAG on CIFAR10 and CIFAR100 remain limited. For instance, compared to FedAF, FedLAG only achieves a 0.86% improvement on CIFAR10 and a 1.24% improvement on CIFAR100.
> > 3. FedLAG does not reduce communication overhead compared to baselines like FedAvg. Hence, I do not believe it is appropriate to label the method as “communication-efficient.”
> > 4. Since FedLAG is a PFL method, reporting its performance improvements directly against FedAvg is unfair. Moreover, even on the CIFAR10 and CIFAR100 datasets, the improvements of FedLAG over the best-performing methods are less than 1%. Therefore, I consider the claim “with improvements ranging from an average of 5–7% to 15%” to be an overstatement.
> >
> > In summary, given that the authors did not adequately address my concerns in the rebuttal, I have decided to keep my score.

---

### Official Review · Reviewer_YtVL · 2024-11-03

**Soundness:** 2
**Presentation:** 2
**Contribution:** 2
**Rating:** 5
**Confidence:** 3

**Summary:**

This paper focuses on enhancing model performance in personalized federated learning (FL).  Recent pFL's methods usually face the problem of selecting which layers of the neural network to be personalized. To this end, the authors propose FedLAG to adaptively personalize layers whose gradients exhibit large conflicts across different clients.  Comprehensive theoretical and empirical results confirm the advantage of the proposed method in terms of accuracy.

**Strengths:**

**S1**. The authors provide a thorough theoretical analysis to show the strength of FedLAG in convergence and model performance.

**S2**.  Experimental results show that FedLAG outperforms baselines in terms of accuracy and convergence time, especially under high data heterogeneity and low user participation.

**S3**. The proposed method is simple bug effective and can be easily integrated into existing methods.

**Weaknesses:**

**W1** . It is quite weird that the modern neural network ResNet9 can only achieve at most 97% testing accuracy on MNIST across all the methods and some results are even lower than 90% in Table 2 (e.g., PerAvg and FedPAC). A simple LeNet-5 model can already achieve 90% testing accuracy on MNIST by FedAvg even in the case of extremely non-i.i.d. partition (e.g., one label per client) [a]. In addition, the performance of FedAvg on MNIST-Dir(0.5) is lower than its performance on MNIST-Dir(0.1), which is also strange since a higher degree of data heterogeneity leads to a better performance for a non-personalized method. These factors make the experimental results less convincing. Please explain the potential reasons for these anomalous results.

**W2**. There may be some typo errors. For example, in the first line of Sec. 7.1.1, the α in the sentence "We assess at two heterogeneity levels, i.e., α = 0.1, 1."  is inconsistent with α 's values in Table 2.  In the Sec. 6.2, the method name GBFL (Zhang et al., 2023c) should be GPFL. Please correct these issues and check over the manuscript.

**W3** Some works related to leveraging gradient conflicts in FL [b][c][d] were missing. More discuss on how the approach relates to or differs from the gradient conflict handling methods is needed. One highly related baseline [e] similarly tackling the layer personalization problem in FL was also ignored. Please compare with this method in the experiments.

[a] Zhao Y, Li M, Lai L, et al. Federated learning with non-iid data[J]. arXiv preprint arXiv:1806.00582, 2018.

[b] Pan Z, Li C, Yu F, et al. FedLF: Layer-Wise Fair Federated Learning[C]//Proceedings of the AAAI Conference on Artificial Intelligence. 2024, 38(13): 14527-14535.

[c] Wang Z, Fan X, Qi J, et al. Federated learning with fair averaging[J]. arXiv preprint arXiv:2104.14937, 2021.

[d] Hu Z, Shaloudegi K, Zhang G, et al. Federated learning meets multi-objective optimization[J]. IEEE Transactions on Network Science and Engineering, 2022, 9(4): 2039-2051.

[e] Ma X, Zhang J, Guo S, et al. Layer-wised model aggregation for personalized federated learning[C]//Proceedings of the IEEE/CVF conference on computer vision and pattern recognition. 2022: 10092-10101.

**Questions:**

Please see the weakness.

---

> ### Author Response · Authors · 2024-11-21
> **Response to Reviewer YtVL 's Weaknesses**
>
> > **Weakness 1:** It is quite weird that the modern neural network ResNet9 can only achieve at most 97% testing accuracy on MNIST across all the methods and some results are even lower than 90% in Table 2 (e.g., PerAvg and FedPAC). A simple LeNet-5 model can already achieve 90% testing accuracy on MNIST by FedAvg even in the case of extremely non-i.i.d. partition (e.g., one label per client) [a]. In addition, the performance of FedAvg on MNIST-Dir(0.5) is lower than its performance on MNIST-Dir(0.1), which is also strange since a higher degree of data heterogeneity leads to a better performance for a non-personalized method. These factors make the experimental results less convincing. Please explain the potential reasons for these anomalous results.
>
> In our work, we use the settings from well-known FL benchmark, so-called pFLlib. Currently, there are various SOTA that leveraged pFLlib as benchmarks [R1] as well as other SOTA [R2], [R3], [R4], [R5] achieve the approximately similar results (the accuracy on MNIST achieves the relatively low performance).
>
> According to the non-iid partition, we have made a mistake according to two different tables. We will make a revision in our update manuscript. To validate our algorithm along with other baselines, we also provide parts of our running results which is demonstrated in the anonymous wandb link:  https://wandb.ai/anonymous-anonymous/fedlag
>   * [R1] Canh et al., Personalized Federated Learning with Moreau Envelopes, NIPS 2020.
>   * [R2] Prateek et al., Bayesian Coreset Optimization for Personalized Federated Learning, ICLR 2024.
>   * [R3] Xinmeng Huang et al., Stochastic Controlled Averaging for Federated Learning with Communication Compression, ICLR 2024.
>   * [R4] Wei et al., Understanding Convergence and Generalization in Federated Learning through Feature Learning Theory, ICLR 2024.
>   * [R5] Zhuang et al., FedWon: Triumphing Multi-domain Federated Learning without Normalization, ICLR 2024.
>
> > **Weakness 2:** There may be some typo errors. For example, in the first line of Sec. 7.1.1, the α in the sentence "We assess at two heterogeneity levels, i.e., α = 0.1, 1." is inconsistent with α 's values in Table 2. In the Sec. 6.2, the method name GBFL (Zhang et al., 2023c) should be GPFL. Please correct these issues and check over the manuscript.
>
>   Thank you for pointing out these typographical errors. We will correct the α values in Section 7.1.1 and the method name in Section 6.2, as well as review the manuscript for any other typos. We appreciate your careful review and feedback.
>
> > **Weakness 3:** Some works related to leveraging gradient conflicts in FL [b][c][d] were missing. More discuss on how the approach relates to or differs from the gradient conflict handling methods is needed. One highly related baseline [e] similarly tackling the layer personalization problem in FL was also ignored. Please compare with this method in the experiments.\\
>     [a] Zhao Y, Li M, Lai L, et al. Federated learning with non-iid data[J]. arXiv preprint arXiv:1806.00582, 2018.\\
>     [b] Pan Z, Li C, Yu F, et al. FedLF: Layer-Wise Fair Federated Learning[C]//Proceedings of the AAAI Conference on Artificial Intelligence. 2024, 38(13): 14527-14535.\\
>     [c] Wang Z, Fan X, Qi J, et al. Federated learning with fair averaging[J]. arXiv preprint arXiv:2104.14937, 2021.\\
>     [d] Hu Z, Shaloudegi K, Zhang G, et al. Federated learning meets multi-objective optimization[J]. IEEE Transactions on Network Science and Engineering, 2022, 9(4): 2039-2051.\\
>     [e] Ma X, Zhang J, Guo S, et al. Layer-wised model aggregation for personalized federated learning[C]//Proceedings of the IEEE/CVF conference on computer vision and pattern recognition. 2022: 10092-10101.
>
> We thank the Reviewer for indicating some similar works to our works. Per Reviewer's comment, we will add [b] [c] [d] into our discussion of related works and their difference and currently working on the results of [e]. Besides, want to discuss about [a] and [e].
>
> - [a] is the popular work in FL which indicates a terminology of weight divergence, and also motivated us for investigating gradient divergence. However, [a] did not leverage the gradient divergence and also only provide the analysis while not providing any new method.
> - We conducted an investigation into [e] and identified its key mechanism: employing a hyper-network on the server to determine the set of parameters used for averaging, thereby minimizing the average loss value. However, existing studies demonstrate that achieving the lowest loss value at each round does not necessarily ensure appropriate gradient progression. In our research, we propose FedLAG, which utilizes a well-established rationale to design personalized layers. We believe our approach is distinct from [e] and represents an improvement over it.

---

> ### Comment · Reviewer_YtVL · 2024-11-26
>
> Thank you for the detailed response. The response has partially solved my concerns: 1) The low model accuracy on MNIST is attributed to using the popular FL framework pFLlib. However, the response did not directly discuss the reason for this phenomenon.
> 2) Although [e] did not address the gradient consistency problem as described in the response, [e] offers a learning-based mechanism to ease the complex selection of layers to be personalized as depicted in "existing methods require extensive fine-tuning to determine which layers should be used for global aggregation and personalization". Since the authors claimed that the gradient-conflict-based mechanism has the advantage of adaptively determining layers to be personalized, it's required to compare the proposed method against other layer-wise personalization mechanisms. I will keep my rating if there is no further clarification.

---

### Official Review · Reviewer_zT2S · 2024-11-04

**Soundness:** 2
**Presentation:** 1
**Contribution:** 2
**Rating:** 3
**Confidence:** 4

**Summary:**

This paper presents a layer disentanglement mechanism to achieve personalized federated learning. Specifically, the paper distinguishes between personal layers and global layers by examining the conflicts in local model gradients. Consequently, the model can adaptively separate the model parameters into personal and global components. Experimental evaluations demonstrate performance improvements compared to baseline models.

**Strengths:**

S1: This paper focuses on an important issue in the federated learning, i.e., build layer-wise personalized federated optimization framework to solve the data heterogeneity challenge.

S2: The idea is meaningful that the system can disentangle the personalized parameters by monitoring the gradient conflict.

S3: Experimental and theoretical analysis demonstrate the effectiveness of the proposed method.

**Weaknesses:**

W1: Compared with common layer-wise personalized federated learning framework, the proposed method has a higher communication overhead. Common layer-wise methods upload partial layers to the server for global aggregation and keep other parameters as personal component locally. However, the proposed method demands clients to upload full model parameters to the server for distinguishing gradient conflict.

W2: The proposed method incurs high computational costs due to the need to calculate the parameter correlations between any two users.

W3: The presentation is poor and there are many confusing clarifications, please refer to section "Questions" for details.

-----------------------------------------------------------------------After Rebuttal------------------------------------------------------------------------

I have acknowledged the authors' response, but the main concerns have not been adequately addressed or explained. Therefore, I will maintain my original rating and recommend that the authors carefully revise the paper in accordance with all reviewers' suggestions.

**Questions:**

Q1: What is the meaning of vertical axis in Figure 2(a)?

Q2: Does the conclusion of Figure 2(b) contradict the textual interpretation in main text?

Q3: Is it necessary to re-compute the conflict layer in each computation round?

Q4: What is the purpose of the experiments in Section 7.2.5?

---

> ### Author Response · Authors · 2024-11-21
> **Response to Reviewer zT2S's Weaknesses**
>
> > **Weakness 1** Compared with common layer-wise personalized federated learning framework, the proposed method has a higher communication overhead. Common layer-wise methods upload partial layers to the server for global aggregation and keep other parameters as personal component locally.
>
>   We respectfully disagree with the reviewer. FedLAG totally no additional communication overheads compared to other federated learning framework. Specifically, FedLAG leverages the model parameters to estimate the local gradients via Eqs. (1) and (2).
>
> > **Weakness 2** However, the proposed method demands clients to upload full model parameters to the server for distinguishing gradient conflict.
>
> Uploading full model parameters from clients to the server is standard in current FL practices. Consequently, the FedLAG does not violate the FL technical specifications.
>
> > **Weakness 3** The proposed method incurs high computational costs due to the need to calculate the parameter correlations between any two users.
>
>   While the proposed method adds computation on the server side, the gradient conflict score is straightforward to implement and has a low computation cost compared to other metrics. Appendix F.1 demonstrates that FedLAG achieves the approximately similar training time with other researches such as the most simple FedAvg. The more details of training time of FedLAG can be shown in extensive experimental results via anonymous link: https://wandb.ai/anonymous-anonymous/fedlag

---

> ### Author Response · Authors · 2024-11-21
> **Response to Reviewer zT2S Questions**
>
> > **Question 1** What is the meaning of vertical axis in Figure 2(a)?
>
> In Fig. 2a, we illustrate three axes to demonstrate three parameters $w_1$, $w_2$, and $b$ respectively. We acknowledge that the figures lack information of the axes. Thus, we will revise figure accordingly in our revised paper to improve the paper clarity.
>
> > **Question 2** Does the conclusion of Figure 2(b) contradict the textual interpretation in main text?
>
>  We do not think the conclusion of Fig. 2b contradict the textual interpretation in main text. The relevance and significance of Fig. 2b and the textual interpretation of the main text can be considered via three statements:
>
>   1. The current works in model/layer disentanglement mostly fix the number of layers according to the rule: assigning the very first layers as generic layer while the very last layers as personalized layers. The Fig. 2b shows that the rule that layers with high gradient conflicts do not distribute all in the very last layers, thus, the layer disentanglement in current works in model disentanglement in FL is not optimal.
>   2. By considering the gradient conflicts, the generic and personalized layers can be assigned adaptively to align with this phenomenon.
>
>   According to the reviewer’s precious comment, we will revise the section 3 accordingly to provide more information to improve clarity and avoid the misunderstandings for the readers.
>
> > **Question 3** Is it necessary to re-compute the conflict layer in each computation round?
>
>   As we can see from Fig. 2b, the gradient conflicts distribution on each layers vary from very first training rounds to very last training rounds. It means that re-compute the conflict layer guarantee a better generic/personalized layer disentanglement in pFL. According to this comment, we will revise the Section 3 accordingly to explain more carefully to improve the paper clarity and provide more meaningful information.
>
> > **Question 4** What is the purpose of the experiments in Section 7.2.5?
>
>   In most current FL researches, the authors always assume that the pre-trained models (e.g., on ImageNet) are available for transfer learning. The pre-trained models  are always trained with centralized dataset, thus, the models are usually good in performance, and guarantee the FL system robust against weight divergence and achieve better performance.
>
>   However, in practice, the pre-trained models are not always available. By training without pre-trained models, the performance of FL are always degrade significantly and suffer more from weight divergences. By doing the experiments in Section 7.2.5, we can prove that our FL algorithm can achieve better convergence rate when training FL without pre-trained models. This is a very important contribution of our paper and shows our paper’s significance. According to the reviewer’s comment, we will revise and add more explanation in Section 7.2.5 to improve the paper’s clarity.

---

### Note · Authors · 2024-12-20

I have read and agree with the venue's withdrawal policy on behalf of myself and my co-authors.